# Sensing System for Plegic or Paretic Hands Self-Training Motivation [note 1]

**DOI:** 10.3390/s22062414

**Published:** 2022-03-21

**Authors:** Igor Zubrycki, Ewa Prączko-Pawlak, Ilona Dominik

**Affiliations:** 1Institute of Automatic Control, Lodz University of Technology, Stefanowskiego 18, 90-537 Lodz, Poland; w2i21@adm.p.lodz.pl; 2Miejskie Centrum Medyczne im. dr Karola Jonschera, Milionowa 14, 93-113 Lodz, Poland; ewapraczko@poczta.onet.pl

**Keywords:** stroke, stroke rehabilitation, paresis, plegia, wearable device, sensor glove, sensor system, Internet of Medical Things

## Abstract

Patients after stroke with paretic or plegic hands require frequent exercises to promote neuroplasticity and to improve hand joint mobilization. Available devices for hand exercising are intended for persons with some level of hand control or provide continuous passive motion with limited patient involvement. Patients can benefit from self-exercising where they use the other hand to exercise the plegic or paretic one. However, post-stroke neuropsychological complications, apathy, and cognitive impairments such as forgetfulness make regular self-exercising difficult. This paper describes Przypominajka v2—a system intended to support self-exercising, remind about it, and motivate patients. We propose a glove-based device with an on-device machine-learning-based exercise scoring, a tablet-based interface, and a web-based application for therapists. The feasibility of on-device inference and the accuracy of correct exercise classification was evaluated on four healthy participants. Whole system use was described in a case study with a patient with a paretic hand. The anomaly classification has an accuracy of 91.3% and $f_{1}$ value of 91.6% but achieves poorer results for new users (78% and 81%). The case study showed that patients had a positive reaction to exercising with Przypominajka, but there were issues relating to sensor glove: ease of putting on and clarity of instructions. The paper presents a new way in which sensor systems can support the rehabilitation of after-stroke patients with an on-device machine-learning-based classification that can accurately score and contribute to patient motivation.

## 1. Introduction

Stroke is one of the leading causes of permanent disability. Yearly, around 1 out of 1000 persons have a stroke, with over 1.1 million cases in Europe every year [1]. While there are many existing devices and methods for post-stroke physical rehabilitation, patients still experience a lack of motivation, boredom, and loneliness, which negatively influence their rehabilitation outcomes [2]. In this paper, we investigate what should be included in a motivating system for a post-stroke hand rehabilitation during the plegic (paralyzed) or paretic (weakly functioning) hand and propose Przypominajka v2— a glove-based device for reminding patients and motivating them with an on-device machine-learning-based exercise scoring, a tablet-based interface, and a web-based application for their therapists.

Stroke rehabilitation includes different therapies and interventions that can enhance a phenomenon called neuroplasticity. Approximately 80% people after a stroke experienced muscle weakness, impaired exteroceptive sensation or proprioception, aphasia, disturbance of balance and coordination, cognitive dysfunction, and depression [3]. Early and intensive rehabilitation is an important factor in improving motor functions and quality of life of people after stroke [4]. Neuroplasticity is a process that plays a significant part in healing the injured brain [5]. Such factors as repeated movements, the focus of attention, emotions, and reward systems can influence neuroplasticity. These factors can generate synaptic enhancement and lead to the activation of new neural pathways [6]. However, during rehabilitation, therapists usually spend more time exercising the plegic lower limb than the upper one, focusing on the ability to sit up, stand, and walk independently [7]. Therefore, the time of upper limb and hand rehabilitation is limited, and individuals may achieve less functional recovery.

There are glove-based hand rehabilitation devices, such as Raphael Smart Glove, which helps focus patients on their workout using computer-based interfaces [8]. Raphael provides active rehabilitation exercises that mainly focus on patients who already possess some level of control. The Raphael Smart Glove consists of sensors that collect data from the patient’s hand movements and convert them into virtual space movements to create occupation-based hand exercises (such as virtual object grasping, fruit squeezing, etc.).

Focus on motivational aspects can be seen in devices such as MusicGlove, where the device interface allows playing music through functional grips [9]. The device feedback mechanism requires users to train the hand “to the tune” by pressing the thumb to distinct leads on a sensor glove, and is reported to be motivating for the patients participating in trials. However, a hand has to be already highly functioning (Box and Blocks score of at least 7) to start training with the device.

A set consisting of Rutgers Master II-ND haptic glove and CyberGlove was used for hand rehabilitation in research by Adamovich et al. [10]. The devices, along with a web-based monitoring station and database, formed a system used for training finger motion for patients during a chronic post-stroke phase. The training was performed in a virtual environment with four exercises for finger range of motion, speed of movement, strength, and fractionation. The collected data was used to precisely adjust the difficulty levels of the sensorimotor tasks through a target-setting algorithm. The training focused on fingers’ dexterity and velocity, so there was no sensor for a position or orientation tracking of the hand, only the finger flexion and force feedback for the pneumatic actuators. The authors reported improvement in fingers’ dexterity and velocity but stated that focusing only on finger training was a limitation. The authors concluded that further research in exercises for wrist and elbow, patient attention, motivation, and bilateral exercises also utilizing the unaffected hand is needed.

There are continuous-passive-motion (CPM) devices for hands. The Kinetec company sells the Kinetec Maestra series of devices (portable, hand, and wrist) for continuous passive motion of hand and wrist. Kinetec Maestra “Hand” and “Portable” are hand exoskeletons, while “Wrist” is a stationary continuous passive motion device, all commonly used as a part of rehabilitation practice. The device’s manual states clinical benefits such as reducing immobility and possible effusion, preventing joint stiffness, and speeding up the recovery [11]. The devices and other CPM devices intend to move patients’ hands with minimal involvement (passive rehabilitation) in place of manual manipulation by the therapist [12]. However, there is limited patient interaction, and no patient attention is required. There are devices where there is patient involvement, particularly through bilateral exercising. Delden et al. provided a review of 20 such devices [13]. The studies found devices used in acute, subacute, and chronic phases of stroke and were either purely mechanical linkages or robotic devices. Authors conclude that, based on their and other meta-analyses, the bilateral training (in general) appears to be an effective alternative to other forms of treatment and devices can help with increasing training intensity and frequency while the studies up to date do not have statistical power to fully prove their value.

In post-stroke therapy, psychological, cognitive, and motivational aspects are important. Patients after stroke suffer from depression and other neuropsychiatric complications. Depression may make rehabilitation difficult and worsen patient comeback and life quality. In 80% stroke patients, there is a need for antidepressive interventions [14]. Post-stroke apathy, affecting 23–50% of acute stroke patients, is characterized by reduced motivation, lack of initiative, feeling, emotion, and concern. It can overlap with depression but is considered an independent disorder [15].

Patients can also be affected by anosognosia and denial of illness where they do not recognize the deficits in their functioning, or deny them altogether. They also can be overly optimistic and believe that they are already able to move their affected limb while they cannot [15].

Cognitive impairment due to stroke can inhibit patients’ ability to rehabilitate and induce lack of focus, forgetfulness, mind fog, and mood disorders. Patients with cognitive impairments experience worse recovery for activities of daily living [16]. After-stroke patients also suffer from chronic and acute fatigue, with substantial and persistent fatigue reported in 39–76% of after-stroke individuals. Persons with fatigue describe their state as a “sense of weakness, reduced energy …feeling drained of energy…”. Patients use terms such as “tired, weak, exhausted, weary, worn-out” [17].

Luker et al., in the review of stroke survivors’ experiences of physical rehabilitation studies, provide a list of issues and preferences of patients [2]. Patients value physical activity and believe that more is better for recovery. Patients feel bored and alone during an inpatient stay. Particularly, free time was considered unstimulating and negatively influenced mood and motivation. They desire a patient-centered therapy with meaningful tasks and an understandable purpose of therapy. Patients also wanted recreational and social activities through reading materials, games, exercise equipment, and crafts. Independence and autonomy were valued, while lack of control was connected to fear, anxiety, and frustration. Similarly, autonomy was considered a worthwhile goal, and assistance was to be used only when needed. The motivation was considered as something that needed to be nurtured and could be influenced by encouragement and support by hospital staff, other patients, and family. Finally, fatigue influenced other themes, and it was a dominant experience for some.

In our conference paper, we introduced Przypominajka v1 and the concept of patient self-training of plegic/paretic hands with the support of a sensor device. The device in the form of a wearable would remind (przypominaj in Polish) about the training sessions and sense patient motions during them to score patient activity and therefore motivate patients. The device was based on a sensor glove with a wrist-worn control and sensing board. Additionally, a web interface was provided for viewing the session activity [18]. It was based on an Arudino board, with an interface in the form of a small OLED screen and buttons, and an SD card for data recording and direct connection through a WiFi module to the main (therapist) server. The scoring was performed on the device by a simple tree-based classifier. The wearable device was simple, and its main function—reminding about, and motivating during, training bilateral exercising sessions, was evaluated as attractive by therapists. However, preliminary tests and discussions with therapists and patients revealed several issues:The Przypominajka’s v1 user interface was placed over the control box worn on the patient’s wrist. Hand movement could cause patients not to see the instructions and score while exercising.The screen was small (4 cm × 2 cm) and challenging to use for patients with poor sight. Similarly, small buttons were challenging to use for patients.The scoring system was trained and tested on a single person’s data and their validity, while the device was used by other persons needing to be checked.

These issues led to the current study and development of version 2 of the device and this research paper.

Section 2 of this paper presents the Przypominajka v2 design and its components. A system supporting the patient was developed to facilitate self-exercising. Section 2.1 describes design goals and proposed change in user journey compared to a current rehabilitation. Przypominajka creates a more structured exercise environment and enables easier information sharing between patients and therapists after normal therapy hours. Multiple components—sensor glove, tablet-based interface, and servers—form the Przypominajka system, for which the architecture is described in Section 2.2.

Central to the Przypominajka function is the sensor glove, which is an in-house developed device. Its hardware and data acquisition are described in Section 2.3, while Section 2.4 describes filtering and feature extraction. New machine-learning-based scoring based on classification of patients’ data is proposed to provide scoring that is more robust to multiple users. The previously used decision tree model based on two simple features, as well as a new model based on convolutional neural network, are described in Section 2.5. The new model provides better, real-time categorization results while still being implemented on the device.

Patient and therapist interfaces require focus as they are the main interaction points between the system and its users. Its details are described in Section 2.6. A tablet application was developed as a patient interface. The large screen and multimedia capabilities enable a better presentation of exercise instructions. The exercises selected with medical professionals are described in Section 2.7.

We describe and discuss motivational and cognitive aspects of Przypominajka in Section 2.8, of which the main motivational aspect is a scoring mechanism based on exercise classification, described in Section 2.9.

To evaluate the Przypominajka system and understand the scoring system behavior with multiple users, we conducted a study on four healthy participants, described in Section 3. Results, described in Section 4, show high accuracy, precision, and recall for both anomaly and exercise classification but worsening for a new user. A separate case study, described in Section 5, showed that an actual patient with a paretic hand was able to use, and enjoyed using, the Przypominajka v2 system, while some issues regarding ergonomics and quality of instructions were found. This section is followed by the Discussion and Conclusions (Section 6) section, in which the device is compared to others and further work based on current approach limitations is outlined.

## 2. Przypomianajka v2 Design

### 2.1. Design Goals and Constraints

The main idea behind the Przypominajka (“Reminder” in Polish language, Figure 1) is to motivate and remind patients to train their plegic or weakly functioning hand. Patients in such a state do not have enough hand control to use devices such as MusicGlove or Raphael and require a device or technique facilitating active support. Moving and exercising the hand is necessary to enhance neuroplasticity [5] as well as to avoid effects of prolonged joint immobilization, particularly contractures. Passive joint mobilization exercises are used, beginning 24 to 48 h after stroke onset. Such exercises have to be done consistently, ideally multiple times daily [19].

Usually, patients are encouraged to self-exercise with a set of exercises (see Figure 1, proposed by Prof. E. Miller from Jonscher Hospital in Lodz). Such self-training takes place after the other training sessions. A typical session, presented as a user journey, is shown in Figure 2: the patient finishes therapist-assisted training sessions for the day. The therapist reminds the patient about self-training of the hand. The patient performs hand training in intervals. Finally, the patient finishes the training for the day.

There are several issues with such an approach: self-training is unstructured, and patients must choose the exercises based on their memory. As the therapist is not present during the self-training sessions, the correctness and intensity of exercises are not recorded and evaluated. This leads to frequent repetition of some exercises while omitting others, leading to less than full joint mobilization. Patients can forget about the training, particularly when suffering from the cognitive effects of stroke. Patients perform the training alone, and issues such as loneliness and apathy can reduce activity.

Therefore, the goals of Przypominajka’s design were the following:Reduce patient cognitive load through reminding in programmable intervals about the training.Provide clear instructions during the training.Provide a way to record and evaluate patient training both to motivate patient and better facilitate patient–therapist cooperation.Introduce patient, therapists, and family interfaces, as well as data-exchange methods, for guiding self-therapy, motivating patients, controlling therapy, and engaging family in patient activity.

The “user journey” with Przypominajka v2 is shown in Figure 2. The patient finishes the therapist-assisted training. The therapist and the patient agree on a training schedule with Przypominajka. The device reminds the patient to train during specified intervals. Particular exercise is given from a preselected set of exercises. The patient trains the hand with the device using the instruction provided, and score based on patient activity is shown and recorded. The patient is free for a time, after which another session is proposed until the end of training time for the day. The therapist can see the results for the day, discuss them with the patient, and possibly become further motivated.

### 2.2. System Architecture and Components

Przypominajka v2 uses a distributed architecture for sensing, user interfaces, and data management. The elements of the system are shown in Figure 3. The primary sensing part of the device is a sensor glove. The interface for the patient consists of a Tablet application, while patient data is stored on a database managed by the main server. The therapists’ interface is available as a web application connected to the central server.

The sensor glove is responsible for data acquisition, self-management, signal filtering, classifying the data, and communication to the tablet interface.

The tablet interface has a patient application for training and reminding of training sessions and on-device session scheduler, exercise selector, and option to review training sessions.

The raw data and results of particular sessions (scores) are sent to the main server. The server manages data access and records the scores and raw data into a database. The therapists’ interface enables therapists to view the data records and set up the device for a particular user.

In the current implementation, the tablet interface is realized as an Android app created using MIT AppInventor. The main server is a containerized Flask app with a local SQLite database. The therapist’s interface is a containerized StreamLit app.

### 2.3. Sensor Glove Hardware and Data Acquisition

The main hardware element of the sensing hardware is a WEMOS LOLIN32 board, based on an ESP32 microcontroller (Dual Core Tensilica LX6 240 MHz with 520 KB of SRAM and 4 MB of external flash memory). The board has an LTH7R IC-based battery management circuitry to which a 980 mAh battery is connected.

Przypominajka v2 uses a combination of flex sensors and a six-axis IMU to measure the patient’s hand movement. Flex sensors are placed alongside the middle finger (10 cm) and the thumb (14 cm). The placement of the sensors is shown in Figure 1.

Flex sensors by SpectraSymbol are used. These are resistive sensors where sensor bend angle translates to a change in sensor resistance. Sensors display nonlinear change in resistance as a function of bend angle (with a higher sensibility for larger angles); they also exhibit step response decay where a step change in bend angle results in decaying change in resistance [20]. A voltage divider circuit with the flex sensor and a constant value resistor is used, and the output voltage is measured using the microcontroller’s analog to digital converter (ADC). Instead of calibrating the sensors, raw ADC values are used in further processing.

The configuration of voltage divider is shown in Figure 4. The ADC is configured with 11 dB signal attenuation and 12-bits resolution (0.1–3.1 V range, 0–4095 internal).

IMU (integrated accelerometer and gyroscope MPU6080) outputs through I2C interface angular velocity and acceleration. Using the factory-calibrated data, values of acceleration ($a\frac{m}{s^{2}}$) and angular velocity ($g\frac{rad}{s}$) are calculated. The values can be interpreted as related to the movement of the whole hand. Acceleration and rotational velocity in three axes are used as an input value for a machine-learning-based function to calculate the score with further data processing.

### 2.4. Data Filtering and Feature Extraction

Using the acquired raw IMU and flex sensor data we form additional features, later used in user data classification.

Based on the IMU data, the roll and pitch angles are calculated using a (discrete) complementary filter with update rule of the form (Equation (Equation 1)):(1)$$\begin{array}{c} \Delta T=time[n]-time[n-1]\\ roll_{raw}\left[n\right]=atan2\left(a_{y},a_{z}\right)\\ pitch_{raw}\left[n\right]=atan2\left(-a_{x},\sqrt{a_{y}^{2}+a_{z}^{2}}\right)\\ roll\left[n\right]=A*\left(roll\left[n-1\right]+g_{x}\left[n\right]*\Delta T\right)+\left(1-A\right)*roll_{raw}\left[n\right]\\ pitch\left[n\right]=A*\left(pitch\left[n-1\right]+g_{y}*\Delta T\right)+\left(1-A\right)*pitch_{raw}\left[n\right] \end{array}$$
where $a_{x\ldots z}$ is acceleration in $m/s^{2}$, $g_{x\ldots y}$ is angular velocity in rad/s, *A* is a cutoff time constant, and ΔT is time between two consecutive sensor readouts.

The (absolute) yaw angle (hand “heading”) can not be accurately estimated using this method as the rotation axis is parallel to Earth’s gravitational field, so different yaw values would have the same influence on the accelerometer readings. Another sensor such as a magnetometer would be needed for this purpose [21]. However, the absolute yaw angle should have little value in assessing the correctness of the exercise, as any heading is possibly correct while rotational velocity around the z-axis is already included in the feature set.

Sensor readings and calculated angles are concatenated to form a vector v (Equation (Equation 2)):(2)$$v[n]=[a_{x}\left[n\right],a_{y}\left[n\right],a_{z}\left[n\right],f_{1}\left[n\right],f_{2}\left[n\right],g_{x}\left[n\right],g_{y}\left[n\right],g_{z}\left[n\right],roll\left[n\right],pitch\left[n\right]]$$
with $a_{x},a_{y},\;\mathrm{and}\;a_{z}$ being acceleration in the *x*, *y*, and *z* axes, $f_{1},f_{2}$ being the values recorded from the flex sensor (thumb and middle finger flex sensors), *roll pitch* being angle calculated using the complementary filter with update rule shown in (Equation (Equation 1)), and *time* being time in ms (from device start).

The vector v is used in classification using the deep learning model. The vector concatenated with an (anomaly classification) score is later sent to the tablet application.
(3)$$v_{history}\left[n\right]=\left(v\left[n\right],\left[time\left[n\right],anomalyclass\left[n\right]\right]\right)$$
where v is a feature vector and anomalyclass is an anomaly classification score (0–1) as described in Section 2.5.3.

In our previous paper, we also introduced two criteria (Sacc: summed $L_{1}$ norm of acceleration values for a time period and Sflex: sum of smoothed differentials of flex sensor values for a selected time period) that could be used as anomaly classification features and could be calculated on a simple microcontroller such as Arduino’s Atmega328 [18].

For each selected time period of recorded signal we calculate two criteria, Sacc (sum of Equation (Equation 4)) and Sflex (Equation (Equation 5)):(4)$$Sacc\left(n\right)=\sum_{t=n-N_{s}}^{n}|a_{x}\left|+\right|a_{y}\left|+\right|a_{z}|$$

Sacc is the sum of $L_{1}$ lengths of acceleration vectors for a selected time window. $N_{s}$ is a time window. $a_{x}$, $a_{y}$, and $a_{z}$ are acceleration values in three axes.
(5)$$Sflex\left(n\right)=\sum_{t=n-N_{s}}^{n}|f_{1}\left(t\right)*h\left(t\right)+f_{2}\left(t\right)*h\left(t\right)|=\sum_{t=n-f_{s}}^{n}\left|\left(f_{1}\left(t\right)+f_{2}\left(t\right)\right)*h\left(t\right)\right|$$

Sflex is the sum of filtered (discrete convolution) raw sensor values from flex sensors for a selected time window. The filter used is h=[0.5,0.4,−0.4,−0.5] which is a smoothed difference kernel. The *h* kernel was calculated by scaling by −2 and rounding a differential (convolution with [−1, 1] filter) of a size three Gaussian filter with sigma 0.9; these hyper-parameter values were selected in preliminary experiments. The goal of using this filter was to calculate a smoothed difference of the signal, as a change in the flexion would mean that there was a finger motion with smoothing for reducing the effect of noise.

### 2.5. Data Classification for Acquiring Score

During the patient training using the device, the patient receives real-time feedback based on their own actions using the sensor glove. Particularly, points are received for correctly doing the particular exercise. The feasibility and quality of two classification model types were considered for this purpose:Classification based on the type of the exercise. Patient data is categorized into one of six categories based on the model hand exercises. The categories are “basket”, “wrist flexion–extension”, “wrist and fingers extension”, “rolling”, “hand up–down”, and “hand kneading” and are further described in Table 1. Data that could be categorized into the correct category (exercise currently prescribed to the patient) would mean that the patient is performing the exercise correctly; model outputting decision vector with weights distributed (nearly) equally would mean exercise being performed incorrectly. The model outputting the correct category would mean exercise being performed as prescribed.Explicit anomalous behavior detection. In this approach, a binary classifier directly classifies a particular input data matrix as correct—meaning that the patient exercises in a prescribed way or anomalously, meaning that the exercise was not being performed correctly.

Both of these approaches were investigated, as the explicit anomalous behavior detection is conceptually clearer and has potential for a simple solution (such as a decision-tree-based classifier). The classification of the type of exercises has more potential for future, more sophisticated interfaces, so its feasibility (using the particular sensor configuration) was also evaluated.

Data classification should be performed online during the patient’s exercise for the patient to achieve effective feedback. Short time frames and responsiveness, i.e., short system reaction time to changes n patient behavior, is preferred. Therefore, we investigated time frames of around 1 to 2 s. An example of systems behavior is shown in Figure 5. Based on the sensor values, features are calculated from which machine learning system infers whether exercise is being performed correctly.

#### 2.5.1. A Two-Criteria-Based Anomaly Classification

In our conference paper, we introduced a simple, decision-tree-based anomaly classification using the two features Sflex and Sacc [18] (the features are described in Section 2.4). Both of the features are combined using a decision tree, shown in Figure 6, to categorize the patient exercise data. The tree is trained using ScikitLearn 1.0.2 (Class DecisionTreeClassifier) implementation [22]. The max depth of the tree was set to three. Gini criterion was used to measure the quality of a split, with the “best” splitting strategy.

#### 2.5.2. Convolutional-Neural-Network-Based Category Classification

This is used for a data *M* matrix of size $N_{s}\times 10$ where row *i* has a form $M[i,:]=v[n-N_{s}+i]$. It is formed from the last $N_{s}$ vectors v representing sensor data and angles. A convolutional neural network with architecture shown in Figure 7 is used to produce an output vector y of size 6.

Each column of the matrix is first preprocessed using a MinMax Scaler created using training data. For $x_{min}$ being the minimum value for all recorded data from a particular feature *x* in the training dataset and $x_{max}$ being the maximum value, we calculate $x_{scaled}$ using Formula (Equation 6) [23]:(6)$$x_{scaled}=\left(x-x_{min}\right)/\left(x_{max}-x_{min}\right)$$

We form a matrix $M_{scaled}$ that is used as an input to a convolutional neural network.

#### 2.5.3. Anomaly Classification Based on a Convolutional Neural Network

On-device anomaly classification can be conducted using a convolutional neural network similar to the one used in categorizing the exercise type. The last layer of the network is replaced with a dense layer with a sigmoid activation (the structure of the network is shown in Appendix A in Figure A1).

The input is the same data matrix $M_{scaled}$ as in exercise classification, while the output is a float scalar *y* between 0 (anomaly) and 1 (correct exercise). During training, the network can be initialized with weights learned from classification tasks for better learning performance.

#### 2.5.4. On-Device Implementation of the Neural Network

Deep learning models were implemented and trained using TensorFlow 2.6. The models were converted to TensorFlow Lite format prepared for use with TensorFlow Lite Micro inference framework [24].

We used ESP32 as a microcontroller with Python 3 as a programming language using MicroPython as an interpreter. A custom MicroPython firmware by Michael O’Cleirigh was used that integrates TensorFlow Lite for microcontrollers, ulab (a MicroPython implementation of a NumPy Library) to give access for TensoflowLite from MicroPython [25].

The compiled models (shown in Figure 7 and Figure A1) have size of 11,360 bytes for exercise classification and 10,160 bytes for anomaly classification. The models for time window $N_{s}=21$ samples take up around 34 kB (anomaly) and 35 kB of RAM (exercise categories). Both fit ESP32 memory along with the rest of the data-processing software.

Using multiple runs of code fragments responsible for data processing and communication, we estimated the time required for the operations. Data preparation—data scaling, memory operations, angle (pitch, roll) calculation with complementary filter update takes 529 (97) μs, reading IMU data through I2C takes 1.17 (0.31) ms, ADC input (flex sensors) reading takes 143.5 (2.1) μs.

Neural network calculation on ESP32 takes 6.7 (0.53) ms for anomaly or 8.6 (0.56) s for exercise classification. The Bluetooth Low Energy data transfer (UART) of each data vector takes 9.4 (3.7) ms. The whole data acquisition, processing, and transfer of results take less than 20 ms (around 18.5 ms). It is possible to achieve a 50 Hz sampling/data-update frequency, although a lower frequency was used in the evaluation study.

It is possible to use a model with a larger time window. Models with a time window of $N_{s}=42$ samples take 5 kB RAM memory more while processing time grows to around 9.3 (0.59) ms and 13.2 (4.1) ms for anomaly and exercise classification.

### 2.6. Patient and Therapist Interfaces

The Przypominajka v1 interface consisted of an OLED screen and buttons placed directly on the user’s wrist. However, therapists that we have interviewed pointed out that such placement makes it very difficult to see the screen during the training, while small buttons and small size of the screen make the device difficult to use by persons with poor sight and low dexterity.

Therefore, in the current device version, controls and presentation of instructions for the exercises are carried out using a tablet application placed on a 10-inch Android tablet device. The sensor glove has a Bluetooth Low Energy UART connection to the tablet during the exercise session. The application was built using AppInventor. Screens from a training session with Przypominajka v2 are shown in Figure 8.

An additional element supporting the supervision of the patient’s exercises is an interface for the therapist. The interface enables therapists to access patient data remotely. It allows the specialist to observe the user’s exercise performance level and select which exercises the patient should perform and for how long—shown in Figure 9 is the interface build in Streamlit Python Framework.

### 2.7. Training Exercises

We selected six exercises beneficial for patients with paresis of the upper limb with Prof. Elzbieta Miller from Dr. Charles Jonscher Municipal Hospital in Lodz. The selected exercises and their descriptions are shown in Table 1. Exercises focus on particular muscle groups of the hand and forearm, and particular joints such as wrist or finger joints. Performing such joint mobilization/manipulation exercises has numerous rationales, such as increasing range of motion, decreasing pain, promoting muscle relaxation, increasing muscle strength, improving joint nutrition, and overall promoting neuroplasticity through various systemic physiological effects [26].

### 2.8. Motivational and Cognitive Aspects of the Przypominajka

The Przypominajka enhances hand self-training by motivating patients and reducing their cognitive load. It also improves the therapists’ ability to understand patient progress and guide patients further. The relationship between the patient and the therapists when using Przypominajka is shown in Figure 10. The patient trains and interacts with Przypominajka, receiving direct feedback from the device in the form of a score but also receiving feedback from the therapist based on the recordings of the interaction stored on Przypominajka’s servers. Therapists can analyze the patient data to guide the patient and further personalize the interaction.

The main goal of the Przypominajka system design was to motivate patients while considering their possible post-stroke cognitive issues. This influenced form and function of its multiple parts.

The central aspect of Przypominajka is reminding about exercising. As shown in the interaction flowchart (Figure 11), according to a schedule, the patient is reminded about training in the form of an alarm. This is helpful for patients that could otherwise forget about training. The length of the activity and its frequency can be set together with therapists to fit patient abilities and needs. Additionally, after the training, during a break time, the interface shows the remaining time so that the patient can choose to have other activities. Patients see the instructions before each training session (with the ability to skip). Each training session can have a different exercise, preselected by the therapist.

The second motivational aspect of Przypominajka is the scoring system. The details of the score are described in Section 2.9. During the exercise, patients have real-time feedback so that they have an opportunity to understand whether he performs the exercise correctly and more motivation to continue. In a simple gamification scheme, the patient’s task is to collect a set of (five) stars during the exercise. Each star relates to a particular threshold of a sum of scores given by the scoring algorithm.

Motivation is also provided through the recording of the activity. Patient scores and raw interaction data are saved to the server so that after the exercises the therapist can see patient activity. Therapists can motivate patients to train further or reward them when more effort is seen. In the case of patients with difficulty, the graphs that are shown in the therapists’ interface (see Figure 9) can help with the understanding of patient abilities and improve personalization of the therapy.

Przypominajka shows animated instructions for exercises on the tablet screen (Figure 8). The instructions are shown while the patient prepares to exercise and puts on the sensor glove. The patient can still see the particular key-frames (hand poses) of the exercise during the training. To help the patient remember the exercise, clear instructions about the selected exercise are shown each time after reminding the patient.

The patient interface was designed for ergonomics and clearness. The lettering and buttons are large and contrastive. Screens (Figure 8) have single activity and meaning. Exercises are selected randomly from the pool available to the patient to reduce the monotony of training. Additionally, patients can skip the exercise or postpone it to remain in control and have some flexibility.

### 2.9. Machine Learning-Based Scoring Mechanism

The main motivational aspect of Przypominajka v2 is the real-time scoring of the user activity during the exercise sessions. For each second of the interaction, user actions are classified into a binary value: 1 meaning correct and 0 meaning incorrect, using the anomaly classifiers described in Section 2.5. We calculate a value between 0 and the number of stars using Equation (Equation 7) to show the score in the form of stars (as shown in Figure 12):(7)$$\begin{array}{cc} SCORE(n) & =\sum_{t=N_{s}}^{n}anomalyclass\left(t\right)\\ N\_STARS\_SHOWN(n) & =floor(\frac{num_{stars}}{f_{s}*p*T_{max}}SCORE\left(n\right)) \end{array}$$
where $num_{stars}$ is maximum number of stars (five), $f_{s}$ is sampling rate, $T_{max}$ is a length of a training session in seconds, and anomallyclass(t) is an output of an anomaly-classifying function. An additional parameter *p* = 0.8 is a margin of error. The user will receive a perfect score (five stars) if, during the training, 80% of interaction time actions would be classified as correct. During the training session, consecutive stars will be shown, with the number of stars calculated using N_STARS_SHOWN(n) function.

## 3. Experiments for Evaluating the Przypominajka System

We conducted a series of experiments to understand Przypominajka’s ability and potential in inpatient care. A multi-user, multi-day study was conducted for evaluating Przypominajka’s machine-learning models’ classification quality, during the October–December 2021 period.

The user study with a patient with paretic hand was conducted in December 2021, to understand the performance of the integrated Przypominajka system and collect patient and therapist opinions.

### 3.1. Evaluating the Przypominajka System Classification Performance

The goal of this study was to understand the effectiveness of the sensor setup in providing information for classification and understanding possible limits of using training data for new patients and in new settings. We conducted data collection to evaluate the possible performance of our system in two tasks: classification performance in classifying the type of exercise being executed (six types explained in Table 1) and classifying/distinguishing between good/wrong (anomalous) exercising. Two machine learning models were evaluated: a decision tree with Sflex and Sacc features as inputs, and a model with convolutional neural network (CNN) architecture. Both of these can fit the microcontroller as described in Section 2.5.4.

#### 3.1.1. Data Collection and Preparation

The data were collected from four healthy participants who used the Przypominajka Sensor Glove on their right hands. Participants were all female and between the ages of 20–22. We obtained informed consent from all participants. Data were acquired and processed as described in Section 2.3.

In all, 292 training sessions were recorded, of which 96 were recordings of incorrect (anomalous) exercising. Each recording (file) consisted of around 1 min of training for a chosen exercise, an average of 79 s. A total of 21,733 s of exercises were recorded, of which 12,233 s were of correct exercising and 9500 s were of anomalies.

The whole file was labeled with a particular exercise category (see Table 1) and additionally with an anomaly flag. Participants were asked to place or move their hands in a way that actual patients could use—for example, lay hands on the table or make only small movements to simulate incorrect training. Table A1 in Appendix A shows all such behaviors.

Recorded data frames were divided into fragments with time window $N_{s}$. Each such fragment was labeled with a particular exercise category and/or anomaly (incorrect training) label. Different time windows $N_{s}=10$, $N_{s}=21$, and $N_{s}=42$ were investigated.

For the training deep learning model, from a set of training recordings (separate files were used for training and testing set), a batch was collected and each training matrix in a batch was constructed by randomly sampling a file from a training set from a random starting point. Python custom script written with pandas software library was used for this purpose.

For training a decision tree, input data consisted of consecutive, non-overlapping fragments of each exercise file from a set of training files. For each such fragment, Sacc and Sflex features were calculated.

For testing, overlapping (single sample shifted) fragments of exercise files were used, forming 701,707 pairs (input matrix, output label), and 243,575 pairs representing anomalous exercises. As cross validation was used, all data was used in testing.

#### 3.1.2. Evaluation Method

Two evaluation cases were explored: leave-one-subject-out (LOSO-CV) cross validation, where all data from a single participant were used for testing while all other participants’ data were used for training, and five-fold cross validation, where data from all participants were used in both training and testing sets.

For five-fold cross validation, for each fold, a different 20% of the files were selected for testing while 80% of the files were selected for training. For leave-one-subject out, for each of the four participants, training files for a particular participant were selected for a testing set, while all other files with exercise data were used for training.

CNN models were optimized with Adam optimizer, with a learning rate of 0.001. The binary cross-entropy (log loss) loss function was used for anomaly classification. For the exercise classification, categorical cross-entropy loss was used in training.

TensorFlow 2.6 was used for implementing and training the model. Models were trained on Ubuntu Linux System with 32 GB of RAM and Nvidia 2080Ti Graphics Card.

For the random-tree model for anomaly classification, the scikit-learn implementation of the model was used [23].

An ablation study was also conducted. For both five-fold-CV and LOSO-CV models, selected columns of the data matrix were removed, simulating a simplified glove construction or data processing structure.

The evaluated cases for ablation study were following:Normal: all features were used.Noflex: flex sensor data ($f_{1}$ and $f_{2}$) was removed.Nogyro: rotational velocity features were removed ($f_{x},f_{y},f_{z}$).Noaccel: acceleration values were removed ($a_{x},a_{y},a_{z}$).Noangles: calculated rollpitch angles were removed from the feature vector.Noimu: all data originating from IMU ($g_{x},g_{y},g_{z},a_{x},a_{y},a_{z},roll,pitch$) was removed.

For each model, four scores were calculated using the test data: precision, recall, $f_{1}$, and accuracy.

For anomaly classification task, the scores were calculated using Equations (Equation 8)–(Equation 11):(8)$$PRECISION=\frac{TP}{TP+FP}$$
(9)$$RECALL=\frac{TP}{TP+FN}$$
(10)$$f_{1}=\frac{2*PRECISION*RECALL}{PRECISION+RECALL}$$
(11)$$ACCURACY=\frac{TP+TN}{TP+TN+FP+FN}$$
where TP is the number of true positives, TN is the number of false negatives, FP is the number of false positives, and FN is the number of false negatives.

For the exercise category classification task, a weighted average of the scores were used. The weight was the number of true instances for each label (support).

Confusion matrices were calculated for the exercise classification task. Each matrix cell was normalized by the cardinality of the (true) category (row).

## 4. Results

### 4.1. Machine-Learning-Based Classification

#### 4.1.1. Exercise Classification

Using the proposed convolutional neural network and all the features, we achieved accuracy of 0.923 (std 0.02) for $N_{s}=42$, precision of 0.935 (0.0136), recall of 0.923 (0.02), and $f_{1}$ score of 0.925 (0.02) for five-fold cross validation, while using data from all participants in both training and testing data-set. Figure 13c shows that the time window $N_{s}=42$ gives best results in both five-fold and leave-one-out cross validation experiments. The overall best result for five-fold cross validation was achieved in the ablation study when roll and pitch angle features were omitted: accuracy 0.933 (0.02) and $f_{1}$ of 0.935 (0.0187). Figure 13a shows the results of ablation study for five-fold cross validation. The worst results were achieved when no imu-based features were used (noimu), with accuracy dropping to 0.641 (0.0373).

For leave-one-person-out cross validation, the mean accuracy, for normal case, dropped to 0.632 (0.2), recall 0.632 (0.201), and $f_{1}$ 0.63 (0.193), with significantly larger standard deviation (see Figure 13b).

Confusion matrices (Figure 14) show that the classes hand-kneading, wrist and fingers extension, and wrist flexion–extension have the poorest performance. Hand kneading had the poorest performance for new users and was frequently misclassified as hand up–down.

Detailed results are presented in Table A5 for five-fold cross validation and Table A3 for leave-one-out cross validation.

#### 4.1.2. Anomaly Classification

For anomaly classification in five-fold cross validation, best results were achieved for convolutional-network-based model with accuracy of 0.913 (0.0235)%, precision of 0.901 (0.0962), recall of 0.94 (0.0398), and $f_{1}$ score of 0.916 (0.0401). The tree-based classifier had accuracy of 0.836 (0.00596) and similarly poorer other results (see Table 2 and Figure 15). For leave-one-person-out, all scores dropped for both types of models, with the CNN model achieving poorer accuracy 0.78 (0.107) vs. 0.79 (0.0632) but better $f_{1}$ score 0.814 (0.0756) vs. 0.767 (0.105), (also, see Table 3).

Ablation results (shown in Figure 16 and in Table A2 and Table A4) show similar behavior for all cases other than noimu. The results may suggest that similar learning capability may be achieved through a subset of features; particularly, the removal of flexion sensors would not worsen the results much.

## 5. Case Report

In May of 2019, the patient was admitted to the Department of Neurology in Łódź with massive paresis of the left limbs due to an ischemic stroke. On the same day, she left the General Surgery Department (7–8 May 2019), where she had a planned surgery—the removal of the tumor in the right submandibular region. In the afternoon, the condition of the patient worsened. In the Department of Neurology, the patient was treated with thrombolytic therapy but without improvement. At the end of May 2019, she was discharged.

From July to September 2019, the patient was admitted to the MCM Neurological Rehabilitation Department. Dr. K. Jonscher in Łódź and began the process of comprehensive rehabilitation.

Rehabilitation included kinesiotherapy, physical therapy, occupational therapy, and psychological therapy.

Since the end of the rehabilitation cycle, the patient has improved gait efficiency and quality, while the hand muscle tone has remained. Detailed patient history is shown in Table 4.

In December 2021, the patient took part in the test of the “Przypominajka” device. The Przypominajka was set up to remind every 15 min. All Przypominajka exercises were selected as applicable.

She carried out the test twice for 1 h, followed by an interview. The use of Przypominajka was recorded (see Figure 17 and Appendix A. The patient was able to follow all exercises; however, some additional explanation was needed from the therapist.

During the interview, the patient described the following advantages of the device:The patient enjoyed the “stars” element of the interface and felt motivated to continue exercising so more stars appear.The patient felt motivated by the device reminder to start exercising. She understood to stop other activities and start preparing for exercise sessions.She understood that recording whether and how the patient exercises allows for checking whether the patient performs the tasks entrusted to them.The patient suggested that user registration might help personalize the rehabilitation cycle individually to the patient’s needs.The patient stated that the possibility of choosing or skipping exercises was also valuable.

The patient described the following shortcomings:A patient with spastic paresis would have difficulty putting on the device; help was needed from the therapist (see Figure 17).The material was uncomfortable. The patient also noticed that it was delicate. Sensor position could change during the hand movement.Patient said that some of the instructions were difficult to follow and more description (possibly by attaching an audio file) is needed. Particularly, the patient wanted to know whether a particular exercise should be performed in “365 degrees” (meaning in the air) or while keeping the hand on a table.

## 6. Discussion and Conclusions

In this paper, we have described a Przypominajka v2 sensor-glove-based system for reminding and motivating patients with plegic or paretic hands to self-exercise. The tablet-based interface is capable of reminding them during preprogrammed hours, instructing them about the exercises, and motivating them using a score-based award system. Compared to Przypominajka v1 described in our previous paper [18], the main changes are in the patient interface and an online (during the exercise) scoring system based on a convolutional neural network. The microcontroller was changed from Arduino to a more powerful ESP 32, which enabled us to keep the on-device convolutional neural network inference through the use of TensorFlow Lite.

### 6.1. Comparison of Przypominajka v2 to Przypominajka v1 and Other Devices

Current Przypominajka patient interface functions are moved to a separate tablet application. This allows for a much more ergonomic interface (larger buttons and images, animated instructions) and more sophisticated interaction. Patients and therapists can set the therapy details and can see the score history directly on the tablet, while in the previous version, this could be achieved only through a web interface or an SD card text file. Tablet computers are widely available, and other applications for stroke survivors are available, which could also be loaded to the tablet [27]. Regarding future device use, the change to a tablet interface also gives a better perspective to commercialization and use on a larger scale. The sensor glove which would have to be manufactured would have a smaller bill of materials, and electronics would be simplified. Using a web application for therapists means that therapists have a centralized way to monitor possibly multiple patients and can use their own devices (smartphones, laptops) to access the system. A similar system architecture was used by Adamovitch et al. in a virtual reality-based exercise system for hand rehabilitation in which a web portal was set up for patient-database remote access, while patients used a VR-based application set up on a PC computer [10]. A current version of the system, described by Qiu et al., uses a LeapMotion controller with a series of games with real-time feedback by a virtual therapist while a remote web portal is set up for the clinicians and scientists to collect the data, guide patients, and select adequate exercises [28]. Authors reported high adherence to the training regime and improved patient scores in Fugl–Meyer assessment. In the case of our device, the focus was on patients with lower hand function, but motivational enhancement by scoring, virtual support, and remote therapist access is possible in such a setup.

The signal processing and machine-learning-based inference were kept on the Przypominajka v2 mainboard. This enables the use of other devices connected to the wearable glove to form different, possibly more simplified, or personalized interfaces. The architecture of the system uses the Internet of Medical Things concept [29]. The sensor glove is a machine-learning-enabled remote sensor (with Bluetooth and possibly WiFi connectivity); a tablet app forms an embedded user interface with another web interface for therapists using a remote database and server. This approach has multiple benefits: scalability, dynamicity, and genericity. The sensor glove and its tablet interface can be a part of a larger network of connected devices for post-stroke therapy with a unified user interface for therapists. It also has its typical challenges: reliability, meaning in this case mainly correctness of collected information and instructions. The device has onboard sensor validity checks, but some therapist supervision is required. Due to the lack of actuators, the system is inherently safe. The challenge in scaling the system and providing an internet-accessible interface is safety. The patient rehabilitation data is sensitive and private, so robustness to data leaks and hacker attacks is necessary.

The access to the data can also be extended further, particularly to provide web access for family members. Family can play a significant role in post-stroke rehabilitation, and therapists are recommended to engage patient families [30]. The current web application version is limited mainly to therapist use. A simplified and secured interface for the family (with patient consent) would allow family members to support therapy and motivate patients, for example, through monitoring and encouraging patient progress. Interaction with family members could also limit patients’ loneliness during the hospital stay. In general, by aiming at after-hours patient activity with the Przypominajka, we hope to influence the issues described by Luker et al.: patients’ boredom, loneliness, and need for physical activity, knowledge, and control [2].

Other sensor-glove-based devices that are currently available for rehabilitation, such as Raphael or Music Glove, encourage patients to use and exercise control over the affected limb. This is achieved by a set of engaging games (controlling the direction of a plane by supination and pronation, wiping a table with the hand movement) or challenges (creating music) [8,9]. The main difference in our approach is in the type of exercises and the patient’s condition. In the case of these and other smart-glove-like devices, patients must have some level of control over the hand, which they can further improve by exercising. Additionally, the devices are to be used in (daily) sessions while being assisted by the therapist. In contrast, the Przypominajka is to be used by patients with plegic or paretic hands where the other hand is needed to move the affected limb. The goal of the exercise is also different. Przypominajka offers a set of exercises and helps to perform flexion and extension movements in the elbow, wrist, metacarpophalangeal, and phalangeal joints of a paretic hand and supination and pronation of the forearm and flexion of the arm. All exercises are assisted by an unaffected upper limb. The main aim of Przypominajka is to motivate patients to self-train their affected upper limb. Moreover, selected exercises can enable preserving the full range of motion in exercised joints, can prevent joint contracture and muscle atrophy, can improve blood and lymph circulation, and can decrease spasticity and edema of the paretic hand. Considering that the time of upper limb rehabilitation during standard hospital rehabilitation is usually limited, our additional device can significantly improve the chances of functional recovery of the upper limb after stroke.

There are also other devices intended to be used for patients with paretic or plegic hands providing passive (i.e., where external force actuates the limb) exercises. Continuous passive motion machines can move selected joints (usually wrist or fingers) through a preset range of motion. Hands are actuated through electric or pneumatic actuators with force transmitted through cables, linkages, or through direct-drive setups [31]. The Przypominajka v2 does not have any actuators and requires the patient to self-actuate the hand. This significantly simplifies the construction. Moreover, by actively participating in the exercises and observing their own hand movement, brain neuroplasticity is promoted. However, patient self-training during this phase of after-stroke recovery is difficult, as neuropsychiatric complications and cognitive impairments require special care [15]. Participating in an engaging game or a fast-paced challenge may be too difficult and fatiguing, but reminding about training, giving clear commands, and assigning a score that checks if the patient is training may fit the state of many patients in the subacute phase with plegic/paretic hands.

### 6.2. Scoring System Evaluation

The main motivational element of the system is a score based on a machine-learning model inference. The quality of correctly classifying patient actions may directly influence patient motivation and trust in the system.

The current model has the form of an “anomaly classification” in which we achieved maximum accuracy of 91.3% (precision of 90%, recall of 94%, and $f_{1}$ score of 91%) while using the best CNN model and 83% accuracy for tree-based classifier. For categorization, the model achieves an accuracy of 92%. The result is comparable to other systems for action recognition based on sensor gloves; Ahmed et al., in a review of gesture recognition systems for sign language recognition, found results from 80% to 98% accuracy [32]. The large range may come from the number, type, and quality of sensors, type and size of classifier (large CNN based on transfer learning vs. linear models), and dataset size. There is also a strong influence of the relation of the training set to test set (i.e., whether it was tested and trained on the same person or group of persons).

There was a considerable drop in results, particularly in exercise classification between five-fold and leave-one-out cross validation (92.3% to 63.2%). It suggests high individuality in the way participants performed the exercises. Additional verification of training on only one participant’s data showed that models can achieve near-perfect accuracy of 98.5% for exercise categorization (see Table A6 and Figure A2). In papers reviewed by Ahmed et al., best results were also reported in studies where testing and training were carried out on the same person’s data, such as Tubaiz et al.’s study [33].

Changes in performance for a new user for anomaly classification were smaller, from 91.3% to 78%. For the noacell case, the change was even smaller, from 90.6% to 82%. Therefore, even for new users, the system behaves acceptably. That is, with the additional margin of error scaling parameter on the scoring function (see Equation (Equation 7)), the user can still achieve maximum points (stars) during training even when there will be false negatives. Interestingly, when training and testing on a single user’s data, anomaly classification accuracy was smaller (77.9%) than when training on the general set. Investigation of this phenomenon and possible methods of rapid learning for new users are the next steps of this research.

The processing times and RAM requirements of the inference programs with CNN models proposed on the microcontroller show that it is feasible to have near-real-time (around 10 ms processing time) on-device inference. While during the data gathering, the update loop was set to 21 Hz, in the future, a faster loop of 50 Hz is possible. This would give a sampling/update rate comparable to commercially available devices, such as 5DT (75 Hz) or VMG (10–100 Hz) [34]. While hand exercises for plegic hands are performed slowly, a faster sampling rate would allow registration of faster movements and reduce the chances of possible undersampling issues.

### 6.3. Limitations of the Current Study and Further Steps

There are important limitations to this study and the current solution. As patients self-exercise, it is important to educate them at the beginning about the details of exercises to avoid adverse effects of wrong exercise training. This was also noted by Qiu et al., where patients were instructed directly or remotely before sessions and could ask for help while training remotely [28]. Clear and simple exercise instructions, short and direct commands displayed during use, and graphics containing emotions with supportive texts improve the chances that the patient will conduct exercises properly and motivate them to perform exercises systematically. However, in the case study, the patient had some difficulty in understanding the details of the exercise, and further improvements in this area are needed.

The patient’s condition may limit the usability of the device. Vision impairment (e.g., homonymous hemianopsia) and reduced cognitive functions may cause difficulties with reading information from the screen and problems with understanding instructions to exercise. Severe pains, increased spasticity, and edema in the upper limb after stroke may make it impossible to exercise, similar to lack of ability to sit up unaided or supported (all exercises are preferably performed in a sitting position). Another question that needs to be answered is whether patients with affected dominant upper limb, disturbed superficial and deep sensation, or hemispatial neglect can use the device. We cannot determine without further study to what extent the impairments mentioned above may influence the usage of our device.

In our preliminary test, we observed that patients, such as the person in our case study, had difficulty putting on and taking off Przypominajka on their own. This serious problem needs to be solved before proceeding with further studies.

In conclusion, we present a new way in which sensor systems can support the rehabilitation of after-stroke patients with an on-device machine-learning-based classification that can accurately score and contribute to patient motivation. A further, larger group of post-stroke individuals with well-defined inclusion criteria is needed to test our device to evaluate the effectiveness of Przypominajka in motivating hand training in patients with paretic hands. 

## Figures and Tables

**Figure 1 sensors-22-02414-f001:**
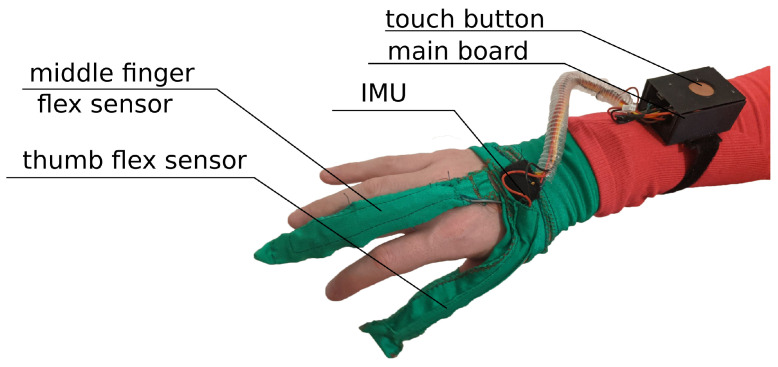
The prototype version of the device.

**Figure 2 sensors-22-02414-f002:**
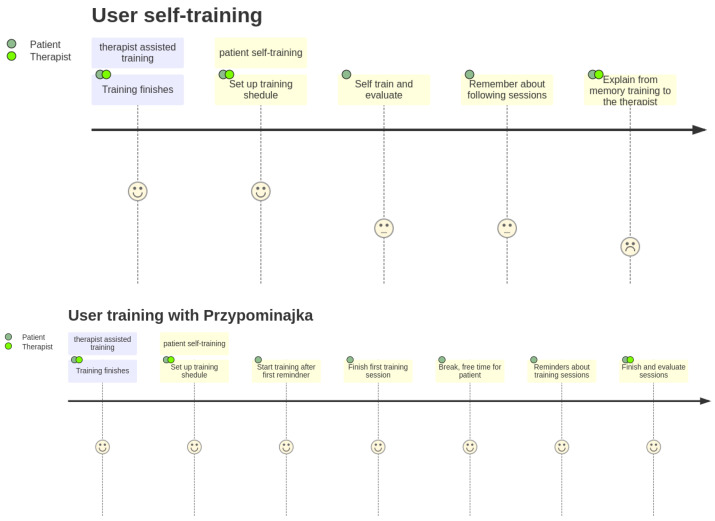
User journeys: with the current self-training scheme and assisted training with Przypominajka device.

**Figure 3 sensors-22-02414-f003:**
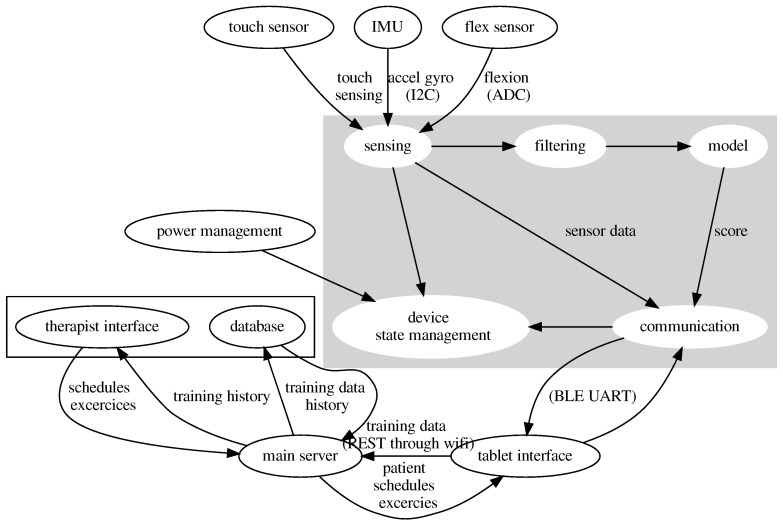
Diagram showing the elements of the Przypominajka v2 system, their role, and communication path.

**Figure 4 sensors-22-02414-f004:**
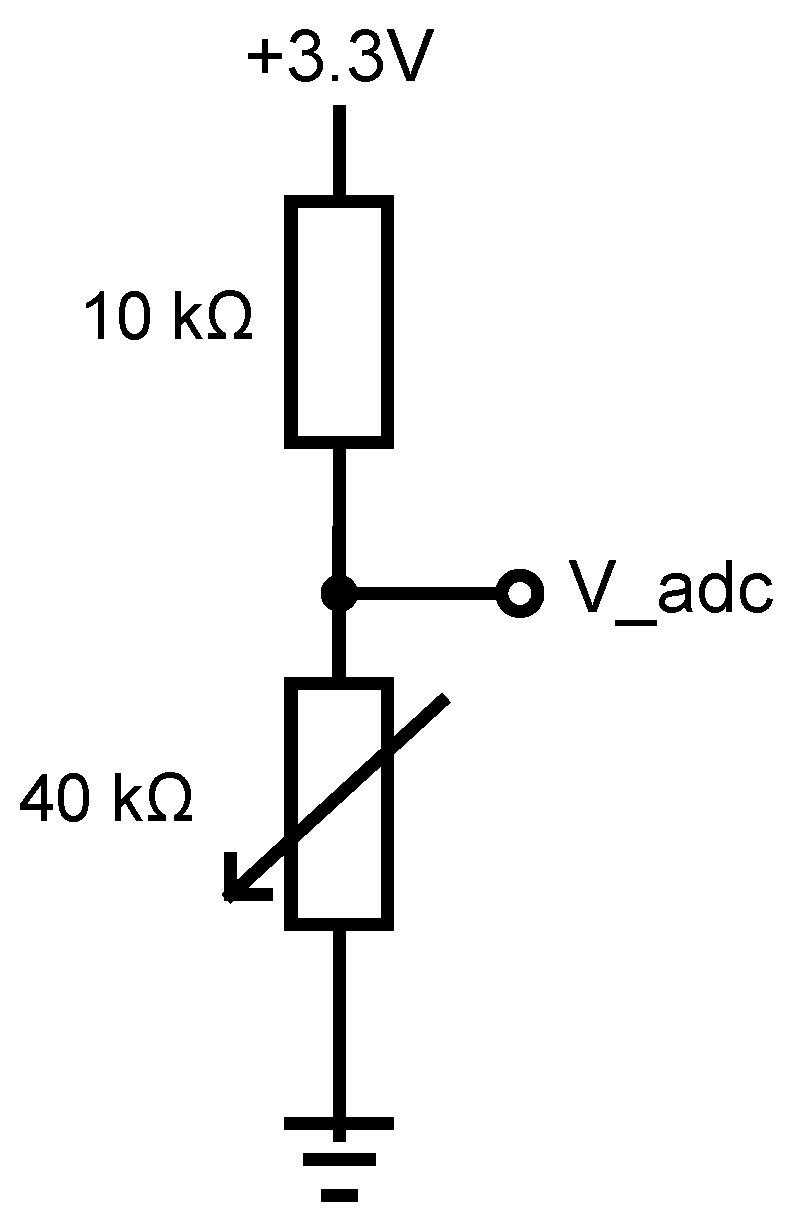
Voltage divider circuit used in measuring the voltage on the flex sensor ($V_{adc}$), represented as a variable resistor.

**Figure 5 sensors-22-02414-f005:**
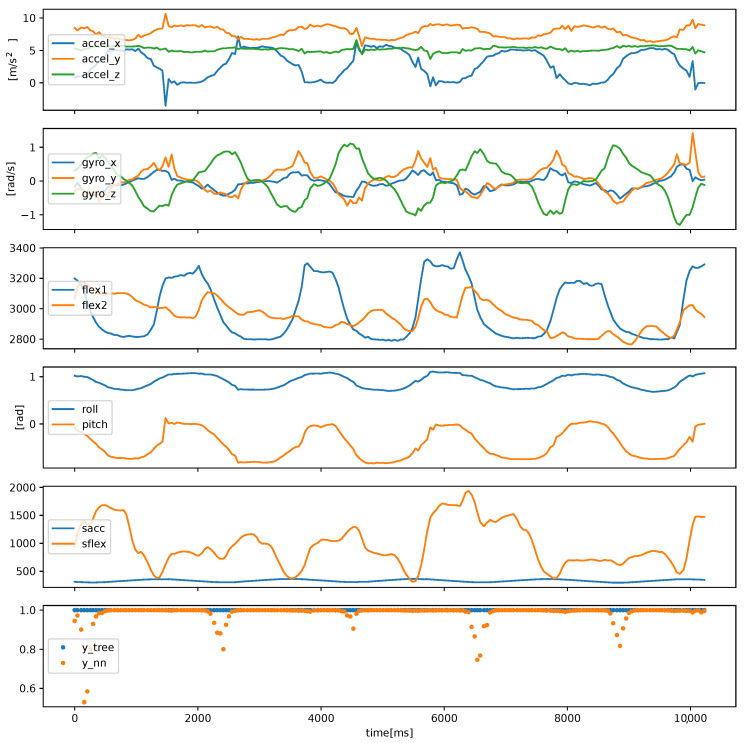
Data series from the sensor glove when the user is exercising correctly. Roll, pitch angles and Sacc, Sflex are computed features. The bottom graph shows the anomaly classifications (anomaly classification scores) using the neural network ($y_{nn}$) and tree-based classifier ($y_{tree}$). Shown is the “wrist and fingers extension” exercise from the test set.

**Figure 6 sensors-22-02414-f006:**
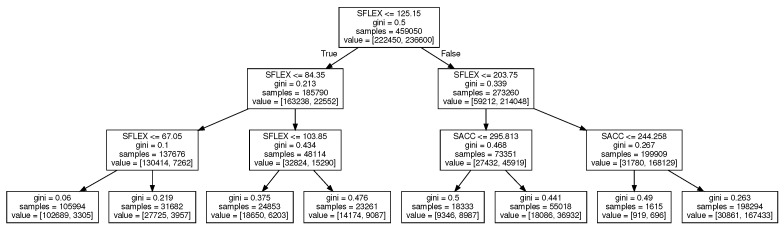
Decision tree that is used on the device to categorize the session. “Gini” means the Gini criterion split quality [22].

**Figure 7 sensors-22-02414-f007:**
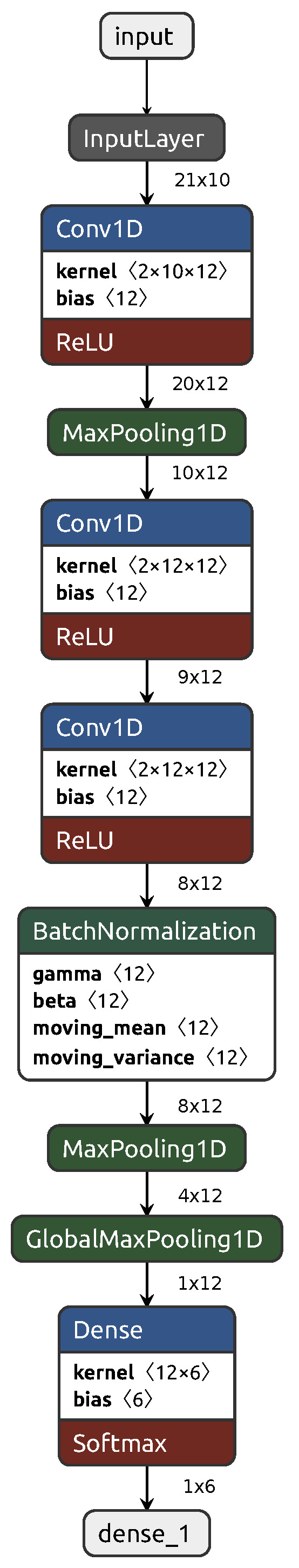
Convolutional neural network for classification of exercise data. During training, categorical cross-entropy loss function was used.

**Figure 8 sensors-22-02414-f008:**
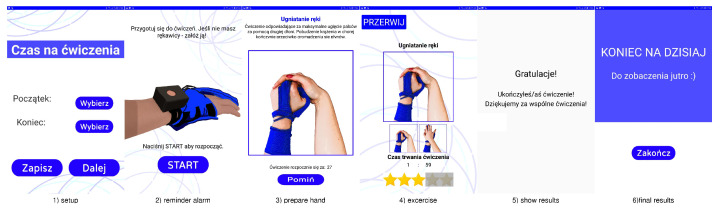
List of screens displayed on the device.

**Figure 9 sensors-22-02414-f009:**
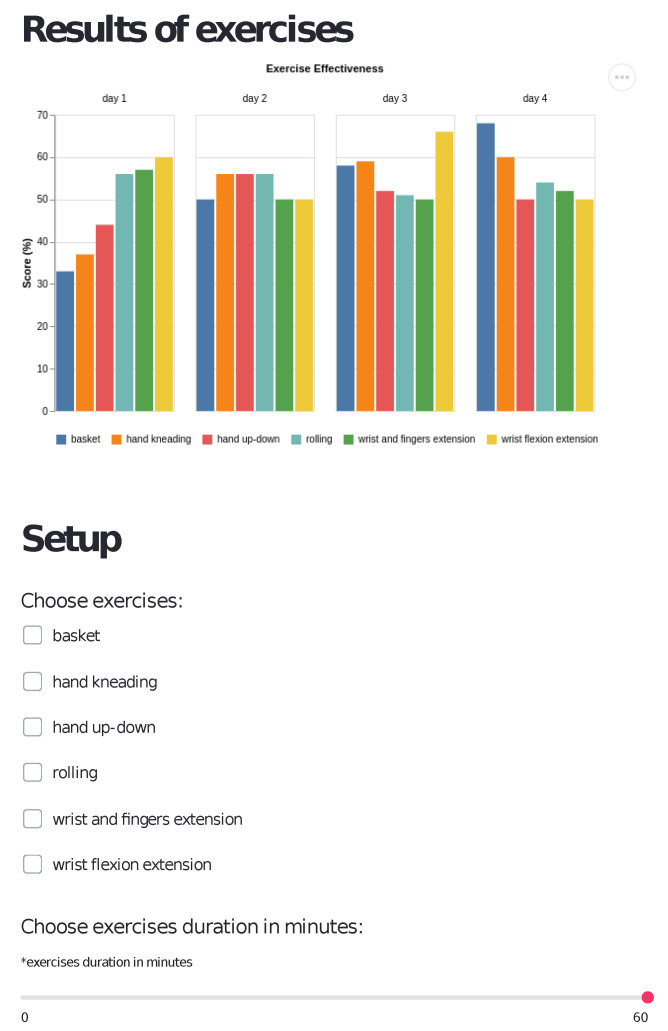
View of the prototype web application for therapists and family.

**Figure 10 sensors-22-02414-f010:**
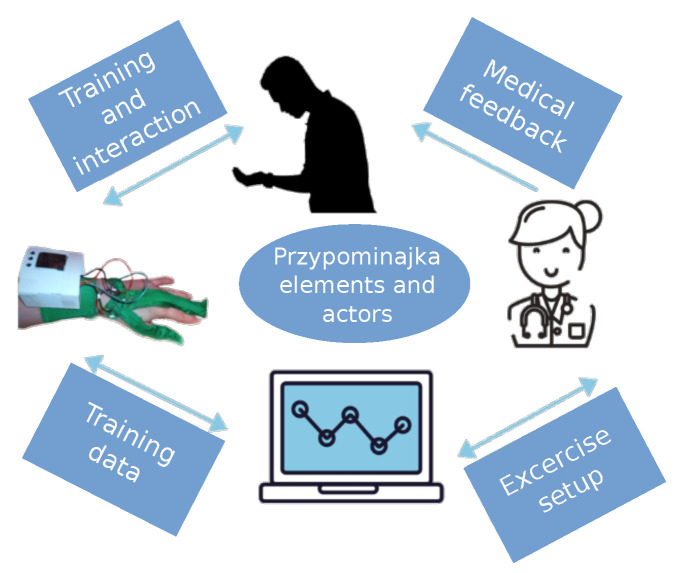
Diagram of the relationship between the patient, device, application, and rehabilitation specialist.

**Figure 11 sensors-22-02414-f011:**
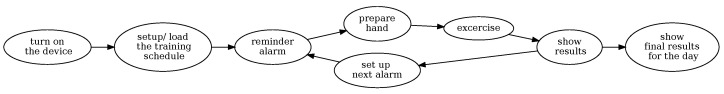
Flowchart of interaction between the patient and the device.

**Figure 12 sensors-22-02414-f012:**
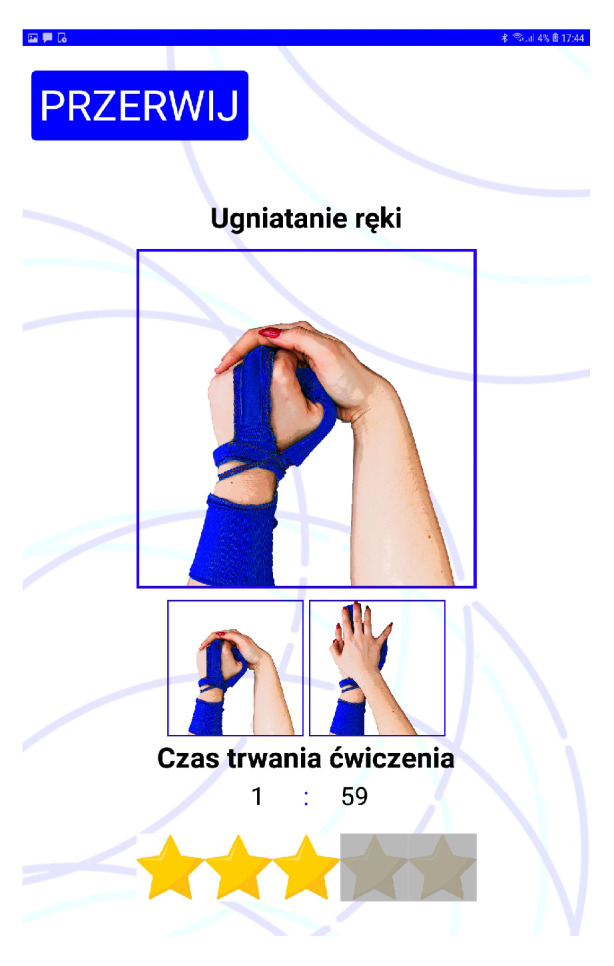
Main motivational aspect of Przypominajka—during the exercise, correct training is awarded with stars.

**Figure 13 sensors-22-02414-f013:**
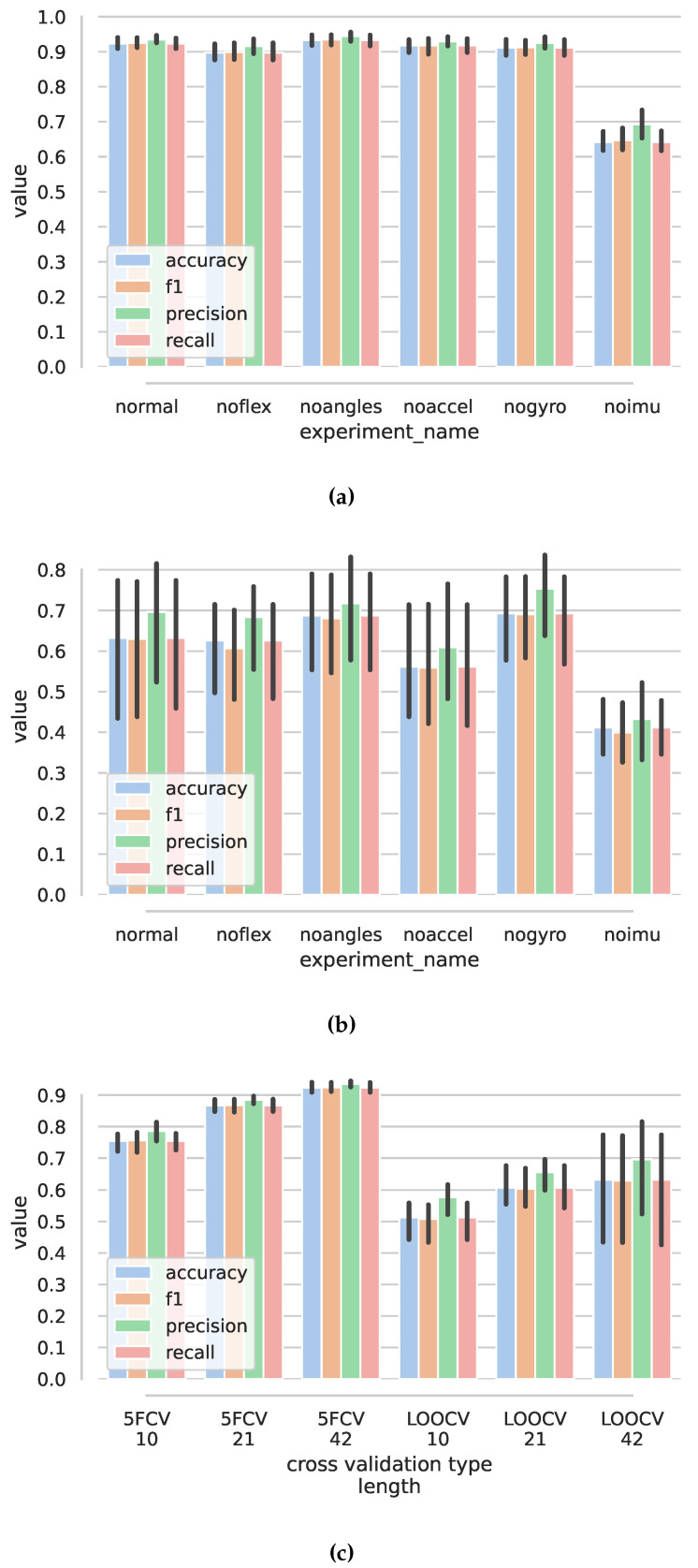
Results for exercise classification using convolutional neural networks. (**a**) Bar plots of exercise classification values for five-fold cross validation. $N_{s}=42$. (**b**) Bar plots of exercise classification values for leave-one-out cross validation. $N_{s}=42$. (**c**) Exercise classification results for five-fold cross validation (5FCV) and leave-one-out cross validation (LOOCV) for different length of time window $N_{s}$ and using all features (normal).

**Figure 14 sensors-22-02414-f014:**
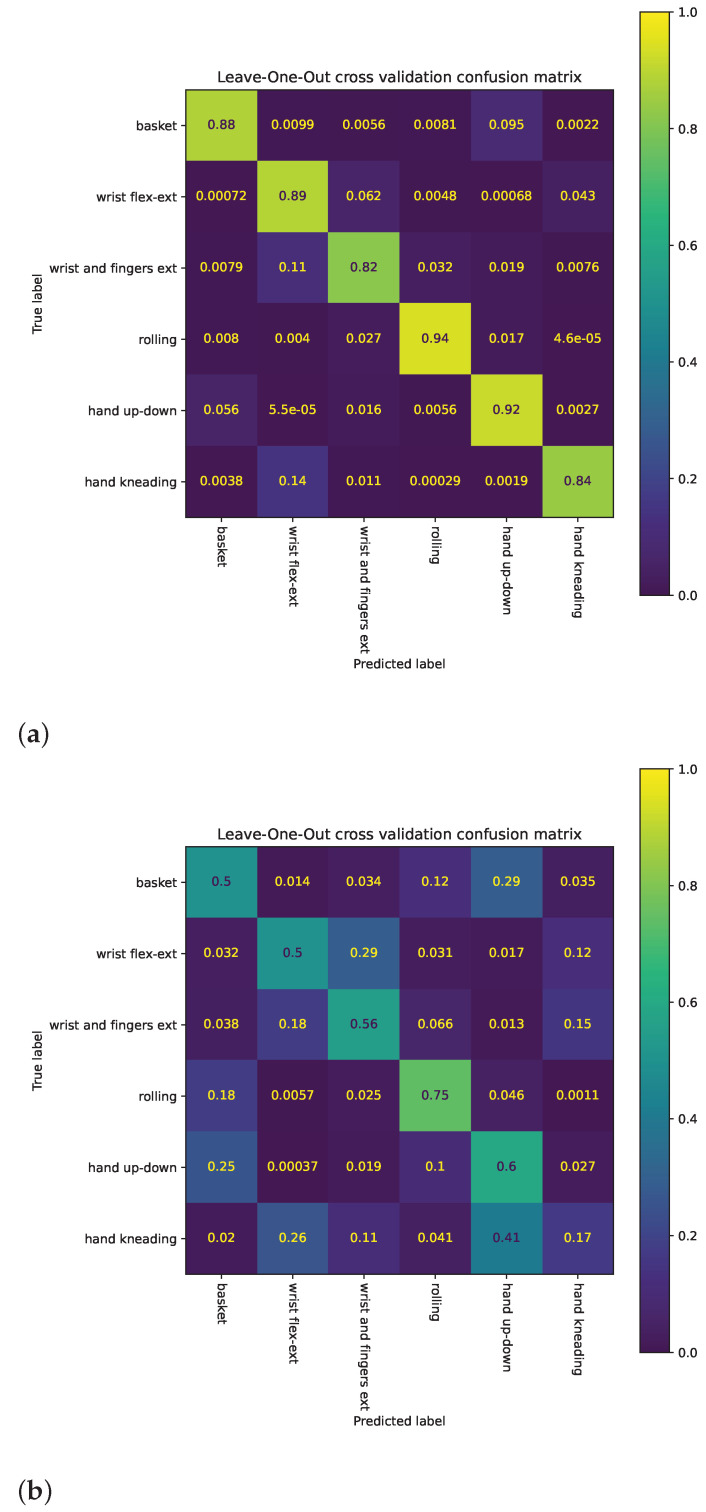
Confusion matrices for classification task (normalized by category cardinality). (**a**) Confusion matrix from five-fold cross validation. (**b**) Confusion matrix from leave-one-subject-out cross validation.

**Figure 15 sensors-22-02414-f015:**
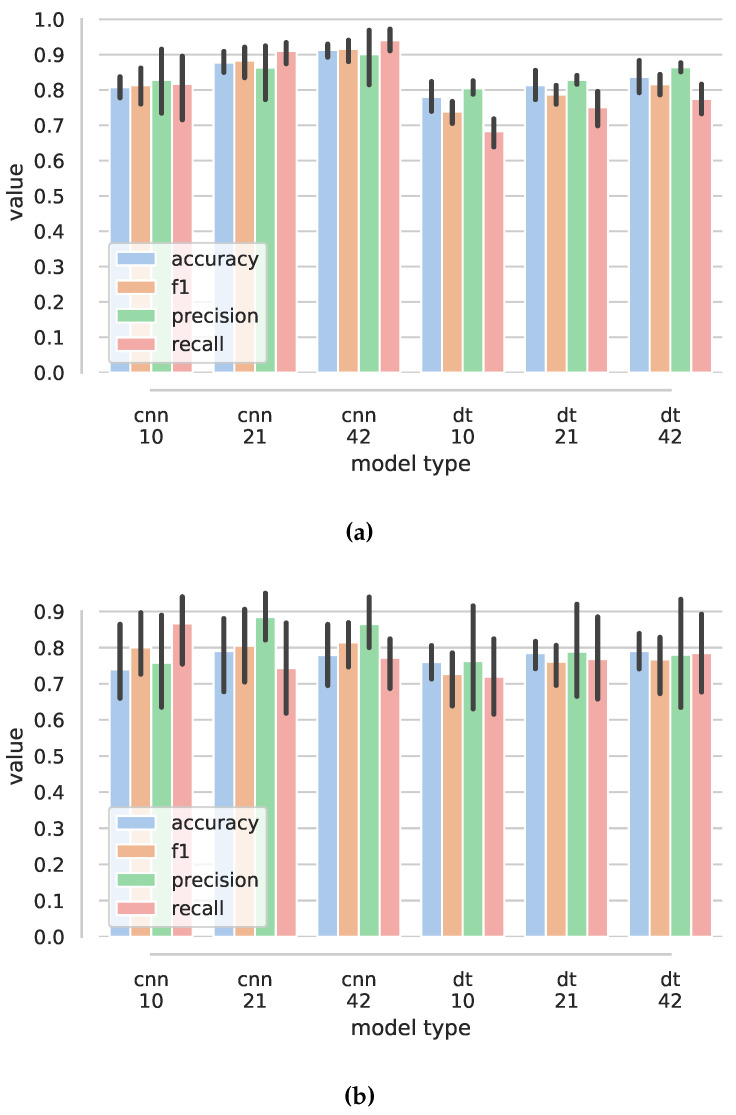
Results for anomaly classification: cnn—convolutional-neural-network-based anomaly classification; dt—decision-tree-based anomaly classification. Numbers below represent the time window length, $N_{s}$. (**a**) Five-fold cross validation results. (**b**) Leave-one-out validation results.

**Figure 16 sensors-22-02414-f016:**
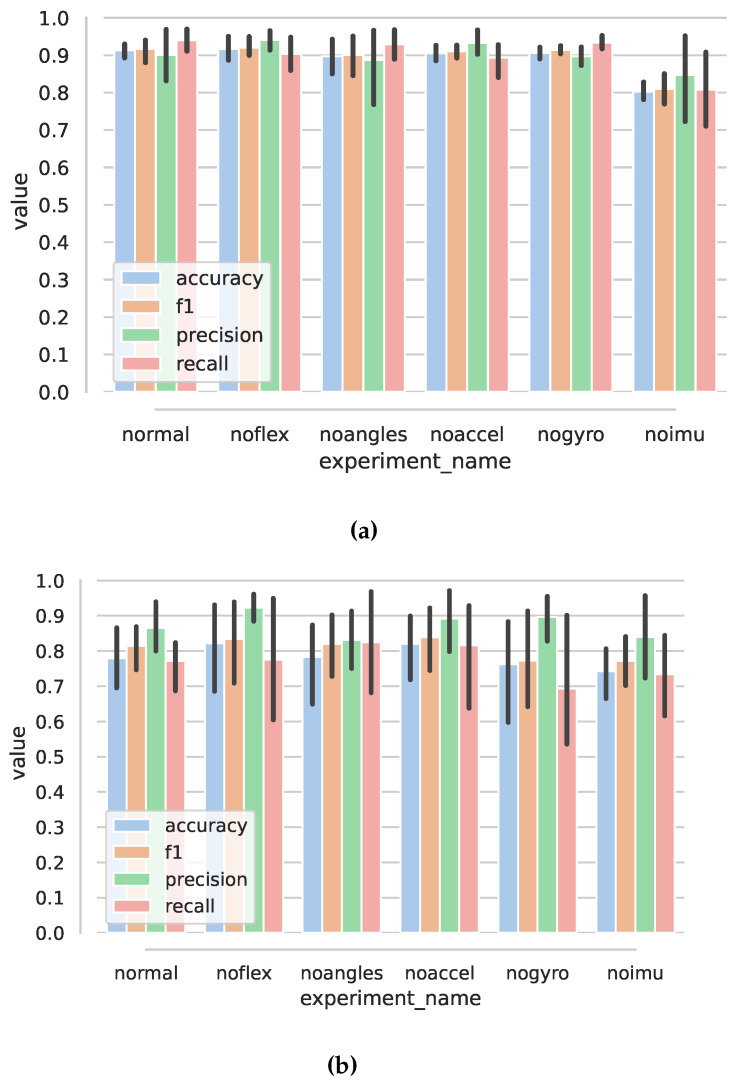
Results for anomaly classification using convolutional neural network. Best case: $N_{s}=42$. (**a**) Bar plots of anomaly classification values for five-fold cross validation. (**b**) Bar plots of anomaly classification values for leave-one-out cross validation.

**Figure 17 sensors-22-02414-f017:**
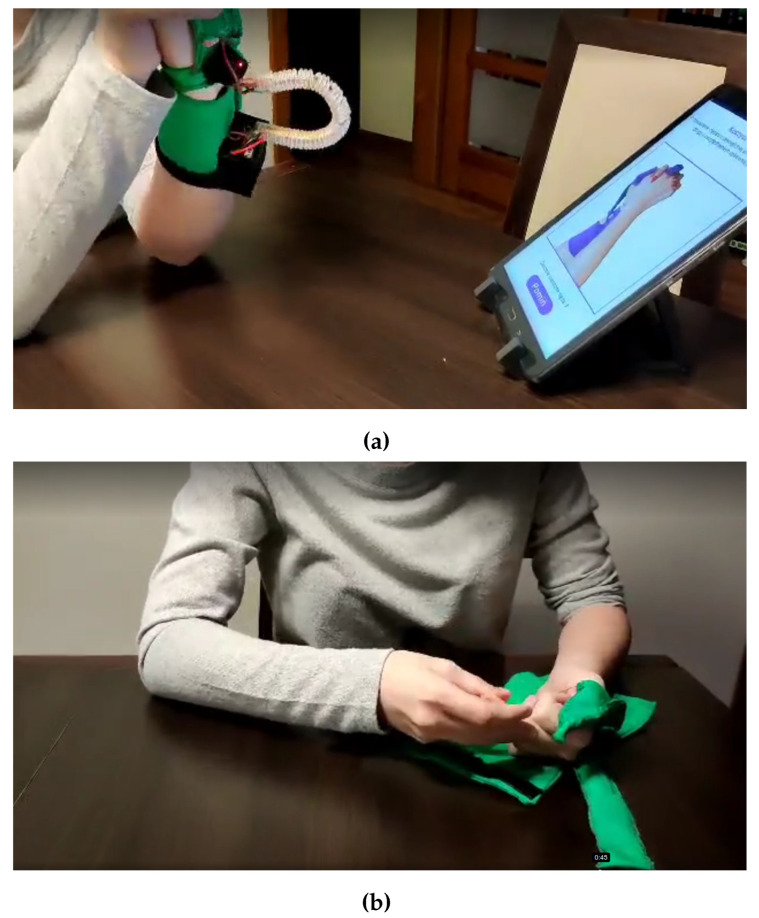
Photographs from a trial session with a Przypominajka user. (**a**) Patient training with Przypominajka v2. (**b**) Patient putting on the Przypominajka v2.

**Table 1 sensors-22-02414-t001:** Joint mobility exercises shown through the Przypominajka interface. The exercises are also classes that the categorizer can recognize.

	Name	Instruction Given
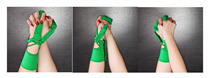	Basket	Bend both your upper limbs in elbows, place your elbows on the table, place your hands and forearms together, bend your fingers in joints, repeatably turn your hands to first see the back of your right hand and then your left one.
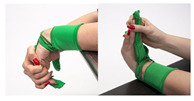	Wrist flexion–extension	Place your affected forearm flat on the table while wrist and hand are off the table, grasp your affected hand below fingers, repeatably bend and straighten in wrist.
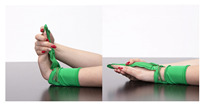	Wrist and fingers extension	Place the affected hand and forearm flat on the table, grasp fingers of the hand; repeatably gently straighten fingers in joints and wrist, then place the hand on the table.
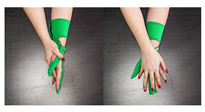	Rolling	Bend both of your upper limbs in elbows, place the elbows on the table, grasp the affected upper limb near the wrist, repeatably turn the affected hand to see the palm—the back of the hand should touch the table, then turn the hand to see the back of it.
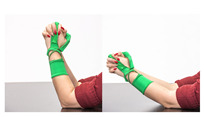	Hand up–down	Bend both of your upper limbs at the elbows, place the elbows on the table, put hands together, repeatably bend the fingers in joints, and bend and straighten upper limbs in elbows.
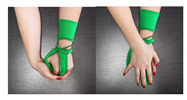	Hand kneading	Place the affected hand and forearm flat on the table, grasp its fingers, repeatably straighten the wrist and bend all fingers, then bend the wrist and put it on the table, then straighten fingers in joints in the maximum range of motion.

**Table 2 sensors-22-02414-t002:** Comparison of five-fold cross validation results for deep-learning-based anomaly classification using all features as input and tree-based classifier.

$N_{s}$	Precision	Recall	$f_{1}$	Accuracy
Convolutional Neural Network				
10	0.829 (0.111)	0.817 (0.117)	0.813 (0.0639)	0.808 (0.0368)
21	0.863 (0.101)	0.911 (0.0398)	0.883 (0.058)	0.878 (0.0374)
42	0.901 (0.0962)	0.94 (0.0398)	0.916 (0.0401)	0.913 (0.0235)
Decision tree				
10	0.804 (0.0239)	0.683 (0.0509)	0.738 (0.0379)	0.78 (0.0541)
21	0.828 (0.0163)	0.751 (0.0609)	0.786 (0.0356)	0.813 (0.0553)
42	0.864 (0.0165)	0.774 (0.0555)	0.816 (0.0372)	0.836 (0.0596)

**Table 3 sensors-22-02414-t003:** Comparison of leave-one-subject-out cross validation results for deep-learning-based anomaly classification using all features as input and tree-based classifier (four persons).

$N_{s}$	Precision	Recall	$f_{1}$	Accuracy
Convolutional Neural Network				
10	0.758 (0.147)	0.867 (0.115)	0.801 (0.106)	0.74 (0.131)
21	0.885 (0.0818)	0.743 (0.149)	0.805 (0.121)	0.79 (0.125)
42	0.865 (0.0867)	0.772 (0.0842)	0.814 (0.0756)	0.78 (0.107)
Decision tree				
10	0.763 (0.179)	0.719 (0.135)	0.726 (0.0956)	0.76 (0.0596)
21	0.788 (0.159)	0.768 (0.148)	0.761 (0.0699)	0.785 (0.0483)
42	0.78 (0.189)	0.785 (0.145)	0.767 (0.105)	0.79 (0.0632)

**Table 4 sensors-22-02414-t004:** Case study patient’s history.

Scale Type (Norm–Max Symptoms)	Neurological Rehabilitation Phase		
	Admission to Hospital	At Hospital Discharge	At Two Years
Barthel ADL (20–0)	12	15	20
Rankin (0–5)	4	4	2/3
Ashworth (0–4)	1+	1+	2

## Data Availability

The data regarding 5-fold cross validation study are openly available in FigShare [10.6084/m9.figshare.19379891], reference number [35]. Other data (LOSO-CV) presented in this study are available on request from the corresponding author. The data are not publicly available due to privacy reasons.

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
