# Peer review of "Sensing System for Plegic or Paretic Hands Self-Training Motivation†"

_sensors, 2022, doi:10.3390/s22062414_

Round 1

Reviewer 1 Report

The architecture of an updated sensing system, especially designed for patients with plegic or paretic hands after stroke, is proposed in this paper. The application field is interesting however the paper fails to present appropriately the proposed methodology and the results. 

Unfortunatelly I cannot recommend it for publication since I have spotted several major deficiencies:

  • The presentation quality is far lower than the one expected. The use of language is very bad and needs to be extensivelly revised. I would say that it needs to be re-written from scratch (the title included).
  • Positioning of the paper is also weakly given in the introduction. The positioning paragraph is weak. It states the target/goal of the study only. What is missing is the means to reach this target. How do they authors reach their targets? Which algorithms / tools were developed to meet the desired goals. 
  • A whole subsection is missing (2.8)
  • Important information is missing e.g. number of recordings performed and collected. From the confusion matrixes, it seems that thousands of training samples have been considered (>20k) but I am not so sure how this was achieved with the six participants that finally participated. 
  • The presentation of results is also poor. For example I dont really understand what the 2 tables (left and right) represent in Table 2 and 3. 

Author Response

We would like to thank the reviewer for the valuable comments.

Referring to particular comments:

The presentation quality is far lower than the one expected. The use of language is very bad and needs to be extensivelly revised. I would say that it needs to be re-written from scratch (the title included).

We reviewed the paper and indeed rewritten its large portions, particularly abstract and discussion.

Considering the reviewer suggestion we changed the title of the paper to „Sensing system for plegic or paretic hands self-training reminding and motivation”

Positioning of the paper is also weakly given in the introduction. The positioning paragraph is weak. It states the target/goal of the study only. What is missing is the means to reach this target. How do they authors reach their targets? Which algorithms / tools were developed to meet the desired goals.

We rewrote the paragraph and abstract to position the paper better. Motivation and description of the approach was added to the end of the introduction section

A whole subsection is missing (2.8)

We returned the missing subsection

Important information is missing e.g. number of recordings performed and collected. From the confusion matrixes, it seems that thousands of training samples have been considered (>20k) but I am not so sure how this was achieved with the six participants that finally participated.

We detailed the missing information. The large size of the dataset is a result of dividing it into overlapping 1 or 2-second fragments.

The presentation of results is also poor. For example I dont really understand what the 2 tables (left and right) represent in Table 2 and 3.

We changed the presentation of the data. Instead of tables, plots are used to better visualize the results.

Reviewer 2 Report

Relatively to the paper called “Sensing system for motivated self-training of plegic or paretic hands“ (manuscript ID sensors-1558488), I believe the paper is well written, providing good contributions in terms of quality and originality, supporting their conclusions with experimental results derived from experimental results and clinical tests. However, some formal issues are found in the manuscript that should be solved for improving its legibility:

  •  
  • The authors are suggested to review the abstract to present the main findings and results of the proposed scientific work, also reporting numerical values.
  • Relatively to the introduction, the authors are advised to further highlight the scope and contribution of the carried out work.
  • The authors are suggested to add a brief overview of the manuscript structure at the end of the introduction.
  • The authors are recommended to review the modality of figures’ citations (not Fig. but Figure) (for further info, see: https://www.mdpi.com/journal/sensors/instructions).
  • The authors are suggested to review the whole manuscript to improve the English language and correct typos. At the current state, the discussion looks quite fragmented.
  • The authors are recommended to clarify all the acronyms at the first appearance (e.g. SACC, SFLEX).
  • The authors are suggested to further discuss the decision tree of Figure 4, along with to substitute Figure 4b with a high-quality version.
  • Relatively to Table 4, the authors are suggested to improve its aspect ratio.
  • Relatively to Figures 9 and 10, the authors are suggested to improve their readability.
  • The authors are suggested to review the citation style of bibliographic references to comply with the journal template (for further info, see: https://www.mdpi.com/journal/sensors/instructions).

Author Response

We would like to thank the reviewer for valuable and detailed comments

Regarding particular issues:

The authors are suggested to review the abstract to present the main findings and results of the proposed scientific work, also reporting numerical values.

We have completely rewritten the abstract to better position the paper and present the main findings

    •  

Relatively to the introduction, the authors are advised to further highlight the scope and contribution of the carried out work. The authors are suggested to add a brief overview of the manuscript structure at the end of the introduction.

We changed the introduction to better present the scope of the paper and outlined the manuscript structure to the motivation behind our approach

    •  

The authors are recommended to review the modality of figures’ citations (not Fig. but Figure) (for further info, see: https://www.mdpi.com/journal/sensors/instructions).

Following the instructions, we changed the figures modality

    •  

The authors are suggested to review the whole manuscript to improve the English language and correct typos. At the current state, the discussion looks quite fragmented.

We revised the whole manuscript and provided a more structured discussion

    •  

The authors are recommended to clarify all the acronyms at the first appearance (e.g. SACC, SFLEX).

We introduced the acronyms (feature names) better

    •  

The authors are suggested to further discuss the decision tree of Figure 4, along with to substitute Figure 4b with a high-quality version.

We provided an extended description of a decision tree and substituted the figure

    •  

Relatively to Table 4, the authors are suggested to improve its aspect ratio.

We changed the Table 4 aspect ratio

    •  

Relatively to Figures 9 and 10, the authors are suggested to improve their readability.

We replaced the Figures with a better quality version and enlarged them

    •  

The authors are suggested to review the citation style of bibliographic references to comply with the journal template (for further info, see: https://www.mdpi.com/journal/sensors/instructions).

We reviewed and improved the references

Once again, we would like to thank the reviewer for valuable comments

Reviewer 3 Report

The manuscript by Zubrycki et al. describes Przypominajka v2 ("Reminder" in the Polish language), which is an integrated sensor system consisting of a wearable device, tablet application, and web application for rehabilitation training and patient motivation. Overall, I like the concept, and the manuscript. The motivation component of rehabilitation is an important issue that the authors are attempting to address. However, there are a few points that this reviewer feels the authors need to address before it is suitable for publication.

  1. The entire manuscript needs to be proof-read again. There are several areas that need corrections (these are some examples):

e.g., line 57: “Patients can also be affected by anosognosia and denial of illness where patients do not recognize the deficits in his own functioning or deny them”. This should be more inclusive, since as written, it only applies to males. Instead of “his”, consider “their”.

e.g.; in line 91 there seems to be a rogue “.”

e.g.; line 89/90 “The device can be used when patient is in bed or while having free time where he is reminded to train her hand using a motivating interface.” grammar. 

e.g.; line 106/107 “Therapists remind the patient about self-training of the hand. The patient does hand training in intervals. Finally Patient finishes” – “P”atient is capitalized when it shouldn’t be.

e.g., line 373:  “Case raport” should read “Case report”.

Table 6 title: “categorizatoin task” should read “categorization task”. Also, why is “categorization” used in Table 6, but “classification” in table 7?

  1. There needs to be more detail and rationale regarding the methods/approach. For example, it seems overly complicated: why does there need to be both categorization and anomaly detection? If one can categorized, then anomaly detection doesn't seem necessary. Also, I assume “data M matrix of size 10XT” refers to a 2-dimensional matrix 10 x 21. Regarding this, why was this window chosen (1 sec at 21 Hz)? Are other window sizes better/worse?
  2. I would prefer to see the CNN architecture schematic presented graphically, with the appropriate 2-dimensional matrices and number of filters and their dimensions illustrated. Also, loss functions?
  3. The computation time required for matrix preparation is noted, but I was wondering about classification/categorization?
  4. Figure 6. In the bar graph, what is “value” referring to; units?
  5. What are the units in the confusion matrix, they don't seem to be percent or error rate.
  6. It would be great for the reader to see some raw data series from the glove and what the case study data looks like in comparison.

Author Response

We would like to thank the Reviewer for valuable and detailed comments.

Regarding the particular comments:

The entire manuscript needs to be proof-read again. There are several areas that need corrections (these are some examples):

We proof-read the entire manuscript and rewritten its large portions

e.g., line 57: “Patients can also be affected by anosognosia and denial of illness where patients do not recognize the deficits in his own functioning or deny them”. This should be more inclusive, since as written, it only applies to males. Instead of “his”, consider “their”.

We changed the pronouns to more inclusive versions

e.g.; in line 91 there seems to be a rogue “.”

We corrected the mistake

e.g.; line 89/90 “The device can be used when patient is in bed or while having free time where he is reminded to train her hand using a motivating interface.” grammar.

We corrected the grammar

e.g.; line 106/107 “Therapists remind the patient about self-training of the hand. The patient does hand training in intervals. Finally Patient finishes” – “P”atient is capitalized when it shouldn’t be.

We removed the capitalization of the word patient throughout the paper

e.g., line 373: “Case raport” should read “Case report”.

We corrected the mistake

Table 6 title: “categorizatoin task” should read “categorization task”. Also, why is “categorization” used in Table 6, but “classification” in table 7?

We corrected the mistake and changed the categorization to classification in all instances

There needs to be more detail and rationale regarding the methods/approach. For example, it seems overly complicated: why does there need to be both categorization and anomaly detection? If one can categorized, then anomaly detection doesn't seem necessary.

We provided more motivation in the section, the main motivation is to understand system’s behavior through the comparison of exercise classification vs the anomaly classification

Also, I assume “data M matrix of size 10XT” refers to a 2-dimensional matrix 10 x 21. Regarding this, why was this window chosen (1 sec at 21 Hz)? Are other window sizes better/worse?

We introduced three different time windows (10, 21, 42) throughout the paper and compared them. The main motivation is that a much longer time window would provide a time delay in feedback, however best classification results were indeed achieved for a longer window

I would prefer to see the CNN architecture schematic presented graphically, with the appropriate 2-dimensional matrices and number of filters and their dimensions illustrated.

Also, loss functions?

We changed the way in which the CNN architecture is presented and added a description of the loss functions

The computation time required for matrix preparation is noted, but I was wondering about classification/categorization?

We described the computation time of all the elements of the on-device processing pipeline in detail

Figure 6. In the bar graph, what is “value” referring to; units?

We changed the graph, the value is the score and it is unitless ( percentages )

What are the units in the confusion matrix, they don't seem to be percent or error rate.

It would be great for the reader to see some raw data series from the glove and what the case study data looks like in comparison.

We changed the confusion matrix to show ratios (row normalized)

We added data series from the interaction with the glove with graphs showing the system classification response

Once again thank you for the detailed comments

Reviewer 4 Report

The paper describes a complex architecture composed of a sensor system, on-device machine-learning-based scoring algorithm, a tablet interface and web application for hand rehabilitation. The system provides a classification and anomaly detection on healthy participants show high precision and recall allowing for accurate scoring and labeling of training activity but with limited effectiveness for new users.

The focus of the paper is interesting and shows a system that could be a relevant application in rehabilitation context. However I have some comments on the content of the article that limit its validity and readability. The sensor system is described with few details. If the sensor system has been described in other papers by the same authors the references should be given. Otherwise more details has to be provided.

Moreover, the on-device machine-learning based scoring algorithm should be given more clearly and with more details. The classes for the classification issues are not clearly defined. Data related to the precision, recall and f1 reported in table 2 and 3 are not clearly described  (what the authors mean in each table for the columns 2-4 and columns 6-8?) . What about accuracy? Why the authors did not report also this parameter?

Details related to the tablet interface and the web application take the great part of the discussion while the considerations about the scoring system are poorly summarized and discussed.

Differences with other systems in discussion is completely missing.

The paper should completely rewritten to increase its readability and to better highlight the novelty issues.

Main concerns

Line 162-The Equation 1 reports, the update for roll and pitch angle. How the authors calculated yaw angles? Do the authors implement a specified algorithm for the evaluation of roll, pitch and yaw angles or they used the onboard timeseries for the angles provided by the IMU? If they implemented a specific algorithm did they have verified the accuracy for the angles timeseries, for instance, comparing their results with those obtained with a gold standard system for kinematic analysis?

Line 165- What fx and fy mean?

Line 170- Typo error, please clarify

Lines 171-178 - The 6 categories are not described and in general the sentences are not clear

Line 179 - Why the authors define SACC and SFLEX as criteria? Can, these two parameters, be considered as features for the following classification? Please clarify.

Line 182 - When the authors say sensor values for flex sensors, do they mean the time series of angles obtained by the 2 sensors? Are then pitch angles or rolls angle?

Please clarify the rationale of the range of values chosen for h

Line 184 - 21Hz is the current sampling frequency for the glove sensors. The sampling frequency is low and able to detect a spectrum movement in a limited frequency range (0-10.5Hz). The authors should discuss this point.

Line 254 - Sentence not clear. Rephrase

Line 261 - Sentence not clear. Rephrase

Line 284 - Numerical indication for the paragraph of this section is misleading. A part is missing?

Line 325 - The authors should indicate for each of the mentioned conditions, the number of features available.

Line 418 - Sentence not clear. Rephrase

Author Response

We would like to thank the reviewer for his detailed and valuable comments. Below we respond to them.

The paper describes a complex architecture composed of a sensor system, on-device machine-learning-based scoring algorithm, a tablet interface and web application for hand rehabilitation. The system provides a classification and anomaly detection on healthy participants show high precision and recall allowing for accurate scoring and labeling of training activity but with limited effectiveness for new users.

The focus of the paper is interesting and shows a system that could be a relevant application in rehabilitation context. However I have some comments on the content of the article that limit its validity and readability. The sensor system is described with few details. If the sensor system has been described in other papers by the same authors the references should be given. Otherwise more details has to be provided.

We provided more detailed information about the sensor system and processing pipeline

Moreover, the on-device machine-learning based scoring algorithm should be given more clearly and with more details. The classes for the classification issues are not clearly defined.

We described the classes, which are in fact the types of exercises that the patient should follow ( 6 classes)

Data related to the precision, recall and f1 reported in table 2 and 3 are not clearly described (what the authors mean in each table for the columns 2-4 and columns 6-8?) . What about accuracy? Why the authors did not report also this parameter?

We added accuracy to all our descriptions. For clarity we added equations for all the measures of the quality. Instead of tables, we added plots for better comparison of the results, while tables were moved to an appendix.

Details related to the tablet interface and the web application take the great part of the discussion while the considerations about the scoring system are poorly summarized and discussed.

We added a section about the scoring system but the lack of evaluation of the scoring system with a group of patients is indeed a limitation of our study

Differences with other systems in discussion is completely missing.

We added comparison to other devices to the discussion.

The paper should completely rewritten to increase its readability and to better highlight the novelty issues.

We have rewritten large portions of the paper (particularly abstract, introduction and discussion) to highlight the motivation and novelty of our approach.

Main concerns

Line 162-The Equation 1 reports, the update for roll and pitch angle. How the authors calculated yaw angles?

We did not calculate the yaw angles only roll and pitch angles were used, we corrected the wording.

Do the authors implement a specified algorithm for the evaluation of roll, pitch and yaw angles or they used the onboard timeseries for the angles provided by the IMU? If they implemented a specific algorithm did they have verified the accuracy for the angles timeseries, for instance, comparing their results with those obtained with a gold standard system for kinematic analysis?

The verification of the angles was not the subject of the study; we consider them as (noisy) features that could be used in classification. Their influence on classification quality was verified through an ablation study and we found that there is little or no improvement when using this feature set compared to a noangle feature combination

Line 165- What fx and fy mean?

These are raw values coming from microcontroller ADC when reading the voltage on the flexion sensor. We improved the description of the features.

Line 170- Typo error, please clarify

We corrected the mistake

Lines 171-178 - The 6 categories are not described and in general the sentences are not clear

We improved the description of categories; these are the types of exercises that the patient should follow.

Line 179 - Why the authors define SACC and SFLEX as criteria? Can, these two parameters, be considered as features for the following classification? Please clarify.

The criteria are defined for legacy reasons as they are part of a decision-tree-based classifier to which we compare the new approach (the features were used in Przypominajka v1). In our opinion, they could be used as a feature for the NN classifier but we decided to use raw sensor values as inputs of the CNN model

Line 182 - When the authors say sensor values for flex sensors, do they mean the time series of angles obtained by the 2 sensors? Are then pitch angles or rolls angle?

The sensor values mean flexion of fingers, the pitch and roll angles are absolute rotations regarding the earth.

Please clarify the rationale of the range of values chosen for h

We added a better description of the filter, which is a rounded differential of a gaussian filter.

Line 184 - 21Hz is the current sampling frequency for the glove sensors. The sampling frequency is low and able to detect a spectrum movement in a limited frequency range (0-10.5Hz). The authors should discuss this point.

We added the discussion and compared the current and maximum sampling frequency (around 50 Hz) to other commercially available devices.

Line 254 - Sentence not clear. Rephrase

Line 261 - Sentence not clear. Rephrase

We rephrased the sentences.

Line 284 - Numerical indication for the paragraph of this section is misleading. A part is missing?

Indeed the whole section was missing, and we added the section back the paper.

Line 325 - The authors should indicate for each of the mentioned conditions, the number of features available.

We provided a better description of all conditions in the ablation study.

Line 418 - Sentence not clear. Rephrase

We rephrased the sentence.

We would once again thank the reviewer for the detailed comments

Round 2

Reviewer 1 Report

The authors addressed my technical comments. The manuscript still needs extensive grammar check and improvements. 

Author Response

Thank you for the review.

We corrected the grammar of the paper as well as provided some further discussion, and improved the readability of the figures.

Reviewer 3 Report

The manuscript has been substantially improved. I have no further comment.

Author Response

We would like to thank the reviewer for the review.

Reviewer 4 Report

The paper was significantly improved and some points not clearly discussed in the previous version are better clarified in the revised version. Several theoretical and experimental aspects are addressed in the paper and this still limits its readability. It is still not clear what is the main aim of the paper given that the authors describe the hardware, the sensor data capture, the model for scoring the performance, the structure of the both the classifications taken into account and finally, they report the case study. The structure of the paper should be re-organized. My suggestion is the consider a section for hardware and data caption description, a section for feature extraction description and the last section for classification. The case study should be inserted as appendix.

Specific comments

  1. There are in the paper some typos that should be corrected. A general linguistic revision should be performed.
  2. Figure 3: some words of the diagram are not clearly readable. The blocks should be described (what is the grey rectangle? What do the black rectangle indicate?).
  3. With respect to IMU, the authors talk about a calibration procedure (line 218). What kind of calibration is performed? What the authors mean with SI (line 218). The authors do not mention the yaw angle. What the authors think about the lack of information related to this measure for describing the hand motion? If the movement of the hand they analyzed, is limited to the roll and pitch they have to add some consideration in discussion.
  4. Figure 8 and figure 13: quality of the image should be improved and the labels of histograms are not easily readable

Author Response

We would like to thank the reviewer for the review.

Referring to the specific comments

It is still not clear what is the main aim of the paper given that the authors describe the hardware, the sensor data capture, the model for scoring the performance, the structure of the both the classifications taken into account and finally, they report the case study. The structure of the paper should be re-organized. My suggestion is the consider a section for hardware and data caption description, a section for feature extraction description and the last section for classification. The case study should be inserted as appendix.

We rearranged the paper according to the suggestions, with separate sections for hardware and data capture, filtering, and feature extraction and classification.

We did not move the case study to the appendix as we consider it important to provide discussion about the device functions and limitations of the current approach

Figure 3: some words of the diagram are not clearly readable. The blocks should be described (what is the grey rectangle? What do the black rectangle indicate?).

We added labels to the diagram’s subclusters and enlarged the figure to improve readability

With respect to IMU, the authors talk about a calibration procedure (line 218). What kind of calibration is performed?What the authors mean with SI (line 218).

We reworded the sentence as it meant that we use factory calibration values. SI meaning standard units of acceleration and velocity, we also reworded the sentence

The authors do not mention the yaw angle. What the authors think about the lack of information related to this measure for describing the hand motion? If the movement of the hand they analyzed, is limited to the roll and pitch they have to add some consideration in discussion.

We added the following explanation to the section: The (absolute) yaw angle (hand "heading") can not be accurately estimated using this method as the rotation axis is parallel to Earth's gravitational field, so different yaw values would have the same influence on the accelerometer readings. Another sensor such as a magnetometer would be needed for this purpose. However, the absolute yaw angle should have little value in assessing the correctness of the exercise as any heading is possibly correct, while rotational velocity around the z-axis is already included in the feature set.

Figure 8 and figure 13: quality of the image should be improved and the labels of histograms are not easily readable

We changed figures to higher (vector) quality and enlarged them

We would once again thank the reviewer for the detailed comments and suggestions